# Alpha-SQL: Zero-Shot Text-to-SQL using Monte Carlo Tree Search

**Boyan Li**[1]  **Jiayi Zhang**[1]  **Ju Fan**[2]  **Yanwei Xu**[3]  **Chong Chen**[3]  **Nan Tang**[1]  **Yuyu Luo**[1]

## Abstract

Text-to-SQL, which enables natural language interaction with databases, serves as a pivotal method across diverse industries. With new, more powerful large language models (LLMs) emerging every few months, fine-tuning has become incredibly costly, labor-intensive, and error-prone. As an alternative, *zero-shot* Text-to-SQL, which leverages the growing knowledge and reasoning capabilities encoded in LLMs without task-specific fine-tuning, presents a promising and more challenging direction. To address this challenge, we propose Alpha-SQL, a novel approach that leverages a Monte Carlo Tree Search (MCTS) framework to iteratively infer SQL construction actions based on partial reasoning states. To enhance the framework's reasoning capabilities, we introduce *LLM-as-Action-Model* to dynamically generate SQL construction *actions* during the MCTS process, steering the search toward more promising SQL queries. Moreover, Alpha-SQL employs a self-supervised reward function to evaluate the quality of candidate SQL queries, ensuring more accurate and efficient query generation. Experimental results show that Alpha-SQL achieves 69.7% execution accuracy on the BIRD development set, using a 32B open-source LLM without fine-tuning. Alpha-SQL outperforms the best previous zero-shot approach based on GPT-4o by 2.5% on the BIRD development set. The code is available at https://github.com/HKUSTDial/Alpha-SQL.

## 1. Introduction

Text-to-SQL (a.k.a. NL2SQL) converts natural language queries into SQL, simplifying access to relational databases and enabling both lay and expert users to derive insights effectively (Liu et al., 2024; Luo et al., 2018a;b; 2022b; 2021; 2022a; 2025; Liu et al., 2025b; Luo et al., 2023). With the advancement of large language models (LLMs), methods like CHASE-SQL (Pourreza et al., 2025) have achieved competitive results on benchmarks such as BIRD (Li et al., 2023c). Generally, these LLM-based Text-to-SQL methods fall into two categories: trained methods and zero-shot methods.

**Training LLMs for Text-to-SQL.** Pre-training or fine-tuning LLMs on task-specific datasets is a common approach to improving Text-to-SQL performance (Li et al., 2024b; Gao et al., 2024b; Talaei et al., 2024; Li et al., 2024a; Zhu et al., 2025). While effective, this method requires extensive labeled datasets and significant computational resources for model training (Lin et al., 2025). Moreover, as newer and more powerful LLMs emerge, the training process must be repeated to maintain competitive performance, further increasing the cost and effort (Wu et al., 2025).

**Zero-Shot LLMs for Text-to-SQL.** As an alternative, *zero-shot Text-to-SQL* methods, such as DAIL-SQL (Gao et al., 2024a) and C3 (Dong et al., 2023), leverage the general knowledge encoded in LLMs to generate SQL queries without requiring task-specific fine-tuning, which eliminates the dependence on labeled datasets and computationally intensive training. While this approach offers a practical and cost-effective solution, it faces a fundamental challenge.

The key challenge in zero-shot Text-to-SQL is the difficulty of transferring and generalizing knowledge from pre-trained LLMs to the specific task of SQL generation, based on natural language queries and database schemas, without fine-tuning on task-specific annotated data. This limitation makes it difficult for the model to handle the complex mapping between natural language queries and diverse database schemas, impeding its ability to accurately interpret schema relationships, construct complex SQL queries, and maintain robustness across various contexts.

**Our Methodology and Contributions.** To address the above challenges, we propose Alpha-SQL, a novel approach that enables zero-shot Text-to-SQL as a process of progressive SQL construction, where queries are progressively built step-by-step. The key idea of Alpha-SQL is to decompose the task into smaller, more manageable sub-tasks, each with

[1]The Hong Kong University of Science and Technology (Guangzhou) [2]Renmin University of China [3]Huawei Technologies Ltd. Correspondence to: Yuyu Luo <yuyuluo@hkust-gz.edu.cn>.

*Proceedings of the 42nd International Conference on Machine Learning*, Vancouver, Canada. PMLR 267, 2025. Copyright 2025 by the author(s).

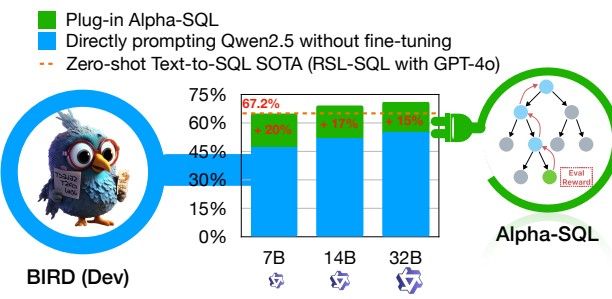

*Figure 1.* Alpha-SQL: A plug-in framework boosting small open-source LLMs. Our method significantly improves Qwen2.5's performance by 15%-20% across different model sizes (7B-32B) without fine-tuning, surpassing even GPT-4o based zero-shot Text-to-SQL SOTA (RSL-SQL) on the BIRD (Dev) dataset.

contextual guidance, making it easier for the model to handle complexity at each step. To achieve this, we model the progressive construction process as a search problem over a tree-structured space, where nodes represent partial reasoning states, and edges denote SQL construction actions (*e.g.,* selecting a table or revising a SQL clause). By iteratively selecting edges (actions) from the root to a leaf node, Alpha-SQL progressively constructs a valid SQL query.

Based on this idea, Alpha-SQL leverages a Monte Carlo Tree Search (MCTS) framework (Coulom, 2006; Xie et al., 2024) to generate and explore SQL construction actions dynamically. To facilitate efficient and effective search within the MCTS framework, we introduce the following novel techniques.

First, to enhance reasoning capabilities during the search process, we propose the *LLM-as-Action-Model*, which invokes an LLM as the reasoning action model in the MCTS framework to generate step-by-step reasoning (*i.e.,* Chain-of-Thought) after each action taken. This reasoning is stored in each node alongside the partial state, enabling Alpha-SQL to maintain context and track the LLM's thought process throughout the SQL construction process. This ensures that each SQL construction action is both context-aware and aligned with the overall reasoning path, which can guide the search toward more promising SQL queries.

Second, to ensure accurate and efficient query generation during the MCTS search process, we introduce a self-supervised reward function to evaluate the quality of candidate SQL queries. Specifically, for each reasoning path, Alpha-SQL generates multiple candidate SQL queries using high-temperature sampling, filters out invalid queries, and computes a self-consistency score by comparing the execution results of the sampled queries with those of the predicted SQL. This helps prioritize promising paths and refines the exploration process. Finally, Alpha-SQL calculates the self-consistency scores of all candidate SQL queries and selects the SQL with the highest score as the final output.

In summary, Alpha-SQL is a fine-tuning-free, plug-and-play Text-to-SQL framework that enhances small open-source LLMs for Text-to-SQL tasks. As shown in Figure 1, it can integrate and boost existing LLMs without fine-tuning on Text-to-SQL datasets. Extensive experiments show Alpha-SQL's strong performance, achieving 69.7% execution accuracy on the BIRD development set, significantly outperforming existing zero-shot methods. Ablation studies confirm the effectiveness of our reasoning actions, and performance improves with more MCTS rollouts.

## 2. Related Work

**Text-to-SQL.** The Natural Language to SQL (Text-to-SQL, a.k.a., NL2SQL) task involves translating natural language questions into SQL queries. The emergence of Pre-trained Language Models (PLMs) like T5 (Raffel et al., 2020) subsequently improved the performance of Text-to-SQL tasks (Scholak et al., 2021; Li et al., 2023a;b; Gu et al., 2023). More recently, the development of LLMs has further advanced Text-to-SQL capabilities. However, applying LLMs directly remains challenging due to issues like schema alignment, complex query generation, etc (Liu et al., 2024). To address this, recent works (Pourreza et al., 2025; Talaei et al., 2024; Li et al., 2024a; Cao et al., 2024; Liu et al., 2025a; Fan et al., 2024) have explored decomposing Text-to-SQL into subtasks, such as candidate SQL generation, refinement, and selection. For example, CHASE-SQL (Pourreza et al., 2025) employs a multi-step pipeline to generate and validate SQL candidates, mitigating errors introduced by direct generation.

Building on this direction, we propose Alpha-SQL, a progressive SQL construction framework that uses MCTS for dynamic query generation. Unlike prior methods relying on static pipelines or fine-tuning, Alpha-SQL leverages LLMs as action models, guiding the search in a context-aware manner, enabling efficient exploration and improved accuracy without task-specific labeled data.

**Test-time Computation.** Recent advances in test-time computation (Snell et al., 2024; Liu et al., 2025a) have significantly improved LLM performance without modifying model parameters. Techniques such as planning, search, and verification during inference have enhanced reasoning across various tasks (Wei et al., 2022; Yao et al., 2023; Qiu et al., 2024; Hao et al., 2023; Zhang et al., 2025; Teng et al., 2025; Zhu et al., 2024; Zhang et al., 2024b). Recent methods, including tree-based search (Yao et al., 2023) and Best-of-N sampling (Qiu et al., 2024), further optimized inference through structured search. Recent work has also explored MCTS-based reasoning to enhance the capabilities of LLMs (Qi et al., 2024). While effective in general reasoning tasks, these methods do not fully address the unique challenges of Text-to-SQL, such as schema understanding,

generating semantically accurate SQL, and refining outputs based on execution feedback.

Alpha-SQL builds on test-time computation principles with a search-based SQL generation framework designed for zero-shot Text-to-SQL. Unlike prior work, it integrates LLM-driven reasoning into the MCTS process for progressive SQL query construction, optimizing the action space and incorporating database feedback to improve accuracy.

## 3. Alpha-SQL

### 3.1. Zero-shot Text-to-SQL

**Text-to-SQL Task.** Let $\mathcal{D} = (\mathcal{T}, \mathcal{C}, \mathcal{R})$ represent a relational database, where $\mathcal{T}$ is the set of tables, $\mathcal{C}$ is the set of columns in those tables, and $\mathcal{R}$ denotes the relationships between tables (e.g., primary key-foreign key constraints). Let $\mathcal{Q}$ denote all well-specified natural language questions over $\mathcal{D}$, and $\mathcal{Y}$ all valid SQL queries over $\mathcal{D}$.

The goal is to find a mapping function $f$ such that for any given question $q \in \mathcal{Q}$, $f(q, \mathcal{D})$ produces a syntactically and semantically correct SQL query $y \in \mathcal{Y}$.

In the **zero-shot setting**, the key challenge is to construct a mapping function $f$ *without task-specific labeled data*. This requires the model to generalize across unseen SQL queries and databases, relying solely on pre-trained knowledge and the provided database schema.

**Formulating Text-to-SQL as a Search Problem.** We define the Text-to-SQL task as a search problem over a vast space of potential SQL queries, where the search space $\mathcal{S}$ consists of all valid SQL queries for a given database $\mathcal{D}$ and question $q$. To structure this space, we represent $\mathcal{S}$ as a tree $\Psi = (V, E)$, where:

(1) **Nodes ($V$)**: Each node $v \in V$ represents a partial reasoning state at a specific step in the query construction process. As shown in Figure 2, The **root node** $v_0$ represents the initial empty query and contains the input question $q$ and database schema $\mathcal{D}$. Intermediate nodes store incremental reasoning steps, such as identifying column values, selecting functions, or constructing SQL clauses. A **leaf node** or **termination node** $v_t$ represents either *a fully constructed SQL query* or *a state where a termination action is applied*.

(2) **Edges ($E$)**: Each edge $e \in E$ corresponds to an *action* in the query construction process, such as selecting a table, adding a condition, or applying an aggregation function. These actions model transitions between intermediate query states in the search tree.

(3) **A Path from Root to Leaf Nodes (Candidate SQL)**: A path from the root node $v_0$ to a leaf node $v_t$ corresponds to a sequence of SQL construction actions that, when composed, forms a complete SQL query $y \in \mathcal{S}$. The SQL query $y$ can

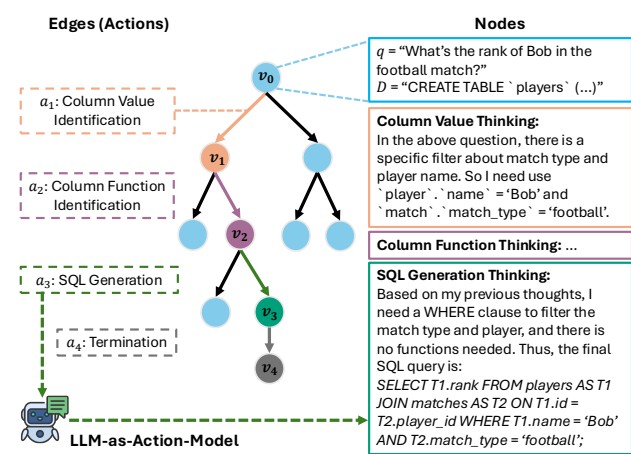

*Figure 2.* Example of the search tree formulation for Text-to-SQL.

be expressed as: $y = v_0 \oplus v_1 \oplus \cdots \oplus v_t$, where $\oplus$ denotes the concatenation or composition of the actions represented by the nodes along the path.

Our goal is to identify an optimal **reasoning path** in the search tree that constructs an accurate SQL query for a given natural language question $q$ and database schema $\mathcal{D}$.

**Complexity of the Search Space.** Efficiently exploring the search space $\mathcal{S}$ for Text-to-SQL generation is a significant challenge due to its combinatorial nature. The size of $\mathcal{S}$, denoted as $|\mathcal{S}|$, grows exponentially with the complexity of the database schema $\mathcal{D}$ and the question $q$. For instance, the number of potential queries grows exponentially with the number of tables, columns, possible join conditions, and nested subqueries. This makes exhaustive search practically impossible, and therefore an efficient search strategy needs to be used to explore this vast space effectively and identify the correct SQL query.

### 3.2. An Overview of Alpha-SQL

To address the challenges of navigating the exponential search space $\mathcal{S}$ and generating high-quality SQL queries from a natural language question $q$, we propose **Alpha-SQL**, a novel framework that leverages Monte Carlo Tree Search (MCTS).

**MCTS-based Search with LLM-as-Action-Model.** Building upon the search problem formulation in Section 3.1, Alpha-SQL employs MCTS to construct and explore the search tree $\Psi = (V, E)$. Given an input question $q$, database $\mathcal{D}$, and an LLM $M$, the MCTS process iteratively builds $\Psi$. The root node $v_0 \in V$ represents the initial state with an empty SQL query. The edges $e \in E$ represent an action $a$, where we invoke the LLM $M$ to select a SQL construction *action* such as schema selection or column value identification, as shown in Figure 2. The MCTS process iteratively

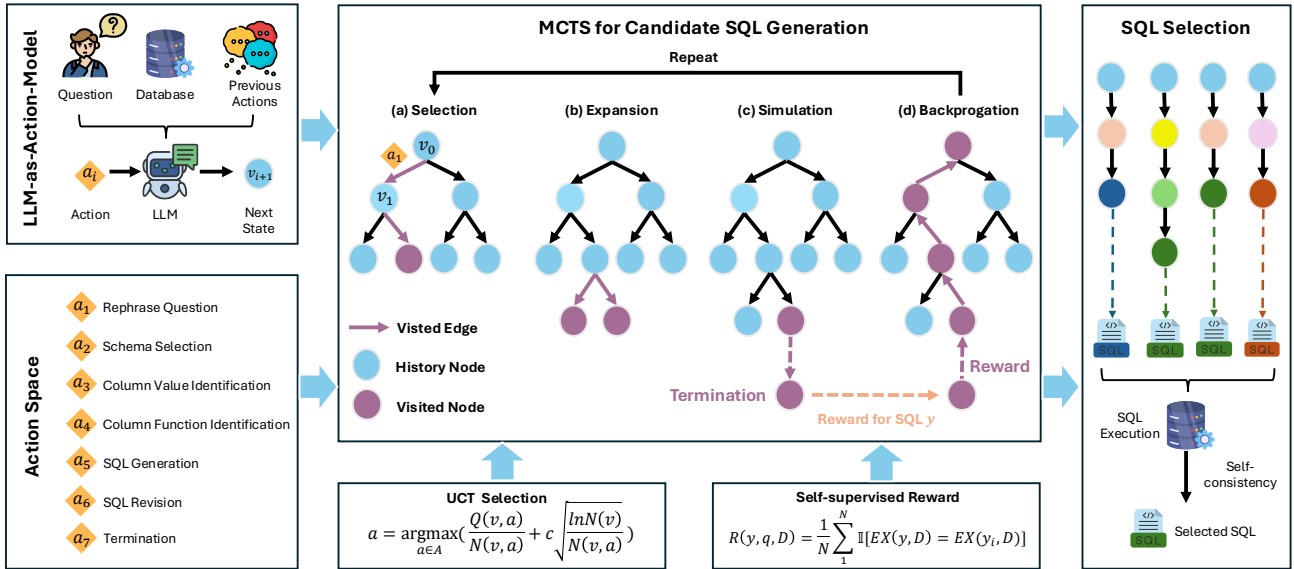

*Figure 3.* An Overview of Alpha-SQL.

invokes $M$ to apply actions during the search, exploring different paths. Each node $v_i \in V$ represents a partial reasoning state after applying a sequence of actions. A complete path from the root to a termination (leaf) node forms a reasoning trajectory corresponding to a candidate SQL query. The MCTS process generates multiple such trajectories, forming a set $T = \{\tau_1, \tau_2, \ldots, \tau_n\}$ of candidate SQL queries. Therefore, Alpha-SQL can efficiently explore the vast search space $\mathcal{S}$ defined in the problem formulation.

**Self-Supervised Rewards.** The reward function plays a crucial role in the MCTS process by evaluating the utility of each action, guiding the search toward more promising SQL queries. Traditional methods like Outcome Reward Models (Zelikman et al., 2022) and Progress Reward Models (Uesato et al., 2022) require domain-specific labeled data for training, making them difficult to generalize across different datasets (Zhang et al., 2024a).

Inspired by human reasoning, we observe that individuals who are confident in their answers tend to consistently provide the same response across multiple attempts, indicating **high confidence** in their solution. Conversely, when responses vary, it suggests **low confidence**, implying uncertainty and lower reliability (Qi et al., 2024). This fundamental intuition directly informs and forms the conceptual foundation of our innovative self-consistency-based reward function. Within this function, the perceived confidence or correctness of a candidate SQL query is quantitatively determined by the consistency of its execution results when compared against those obtained from a diverse set of multiple sampled queries. The reward is computed as:

$$R(y, q, \mathcal{D}) = \frac{1}{N} \sum_{i=1}^{N} \mathbb{1}[\text{Execute}(y, \mathcal{D}) = \text{Execute}(y_i, \mathcal{D})],$$

where $y_i$ are sampled SQL queries, and $N$ is the number of samples. This formulation reinforces SQL queries that consistently yield stable execution results, enabling Alpha-SQL to prioritize reliable reasoning trajectories without requiring annotated data.

## 4. The Design Details of Alpha-SQL

### 4.1. LLM-as-Action-Model

A key challenge in zero-shot text-to-SQL is the difficulty of transferring general knowledge from pre-trained language models to the specific task of SQL generation.

To enhance the reasoning capabilities of our framework, we propose the ***LLM-as-Action-Model***, which leverages LLMs to generate reasoning actions (*i.e.,* CoT Thoughts) dynamically based on the current context of the problem. As shown in Figure 3, LLM-as-Action-Model empowers the LLMs to generate appropriate action outputs based on the question, database schema, and current partial reasoning state (including previous actions), enabling the model to build a valid SQL query in a step-by-step manner.

Formally, at step $i$ of the reasoning process ($v_i$), MCTS selects an SQL construction action $a_i$ from the action space, which is defined later. Based on the reasoning trajectory $v_0 \oplus \cdots \oplus v_i$, the LLM is prompted to execute $a_i$ and generate the next state $v_{i+1}$:

$$v_{i+1} = LLM(q, \mathcal{D}, Actions(v_0, \cdots, v_i), Prompt(a_i)),$$

where $Actions(\cdot)$ refers to all previous reasoning steps in the trajectory, and $Prompt(\cdot)$ represents the prompt instruction for a specific action.

**SQL Construction Reasoning Action Space.** The action space defines the set of potential reasoning steps an LLM can take to decompose and solve Text-to-SQL problems. It is crucial for the LLM-as-Action Model to guide the progressive construction of SQL queries by specifying possible actions at each stage. Previous works (Pourreza et al., 2025; Talaei et al., 2024; Gao et al., 2024b) used limited actions and fixed pipelines, which restricted the model's ability to explore the full space of potential solutions.

Inspired by human thinking, where some might jump straight to the answer while others first clarify the question and break it down into subtasks, we introduce new reasoning actions, such as question rewriting (Qi et al., 2024), alongside existing ones. In total, our action space defines seven distinct reasoning actions. The specific prompts for each action are illustrated in Appendix A.2.

$A_1$: Question Rephrasing. Text-to-SQL systems need to handle diverse question styles and ambiguities from different user groups (Liu et al., 2024). While NL-Rewriter (Ma et al., 2024) addresses this through experience-based question rewriting, it struggles in zero-shot scenarios where training data is unavailable. Building on rStar (Qi et al., 2024), we use few-shot prompting to decompose questions into a structured (conditions list, question) format.

$A_2$: Schema Selection. Databases often contain complex schemas, but individual SQL queries typically use only a small subset of available elements. This mismatch creates challenges for accurate SQL generation (Liu et al., 2024). Prior work has established schema selection as a critical component of SQL generation (Cao et al., 2024; Pourreza et al., 2025; Talaei et al., 2024). Following (Talaei et al., 2024), we use Chain-of-Thought (CoT) prompting to identify the relevant schema subset for each user question, which then guides subsequent query generation.

$A_3$: Column Value Identification. Text-to-SQL systems need to accurately identify filtering conditions in user questions. For example, "What is Bob's best ranking in football matches?" requires filtering on both name ("Bob") and match type ("football") (*e.g.,* `WHERE name = 'Bob' AND match_type = 'football'`). CHESS-SQL (Talaei et al., 2024) found that 20% of errors in the BIRD development set stem from incorrect filtering columns or value selection. To address this, we introduce a column value identification action that evaluates potential filtering values before SQL generation.

$A_4$: Column Function Identification. Complex SQL queries often require aggregate functions (*e.g.,* `COUNT`) and scalar functions (*e.g.,* `STRFTIME`). For example, the question "How many people were born in 2024?" necessitates both date manipulation (`STRFTIME('%Y', people.date_of_birth) = '2024'`) and aggrega-

*Table 1.* Action Space with Ordering.

| Previous Action | Valid Next Actions |
|---|---|
| – | $A_1, A_2, A_3, A_4, A_5$ |
| $A_1$: Question Rephrasing | $A_2, A_3, A_4, A_5$ |
| $A_2$: Schema Selection | $A_3, A_4, A_5$ |
| $A_3$: Column Value Identification | $A_2, A_4, A_5$ |
| $A_4$: Column Function Identification | $A_2, A_3, A_5$ |
| $A_5$: SQL Generation | $A_6, A_7$ |
| $A_6$: SQL Revision | $A_7$ |
| $A_7$: Termination | – |

tion (`COUNT(people.id)`). Analysis by CHASE-SQL (Pourreza et al., 2025) revealed that function-related errors account for 19% of mistakes in the BIRD development set. To improve function handling ability, we introduce a column function identification action during inference.

$A_5$: SQL Generation. SQL generation is the core component of Text-to-SQL systems. CHASE-SQL (Pourreza et al., 2025) introduced a Divide-and-Conquer CoT strategy that breaks down complex queries into multiple subtasks, solves them independently, and combines solutions. This method particularly excels at handling nested queries. We incorporate this strategy into our reasoning action space.

$A_6$: SQL Revision. LLMs can generate syntactically invalid SQL queries in complex scenarios (Wang et al., 2025; Pourreza et al., 2025). Drawing inspiration from human debugging practices, we implement an execution-guided correction mechanism. Our approach provides the LLM with the user question, schema, incorrect SQL, and execution results to guide query revision. The system performs multiple correction rounds until either obtaining a valid SQL query or reaching a maximum attempt limit $N_{revision}$.

$A_7$: Termination. The termination action is invoked when the reasoning process yields the final predicted SQL, signifying the conclusion of a reasoning trajectory. We specify that the termination action must occur following either SQL Generation or SQL Revision actions.

**Action Ordering and Constraints.** Each reasoning trajectory follows a structured order, ensuring logical coherence. For example, some actions, like SQL Revision, must occur only after SQL Generation. Table 1 defines the valid transitions between actions. Furthermore, to prevent the possibility of infinite loops and ensure forward progress in the reasoning trajectory, we impose a constraint that restricts each defined action to appear only once within any single query construction process.

### 4.2. MCTS for Candidate SQL Generation

**Candidate SQL Generation with MCTS Rollout.** We generate Candidate SQL through multiple MCTS rollouts.

Specifically, each rollout includes four distinct phases: *Selection*, *Expansion*, *Simulation*, and *Backpropagation*.

*(1) Selection.* This phase identifies promising nodes for expansion by traversing from the root node ($v_0$) to either an unexpanded leaf or termination node. We use Upper Confidence Bound applied to Trees (UCT) (Kocsis & Szepesvári, 2006) to balance exploration and exploitation during node selection: $UCT(v, a) = \frac{Q(v,a)}{N(v,a)} + c\sqrt{\frac{lnN(v)}{N(v,a)}}$, where $N(v, a)$ counts visits to action $a$ from node $v$, and $N(v)$ tracks total visits to node $v$. $Q(v, a)$ is the estimated reward for action $a$ from node $v$, updated via backpropagation. We select the action with the maximum UCT score and move to the resulting child node. Notably, if there exist any unvisited child nodes (*i.e.*, $N(v, a) = 0$), we prioritize the selection of such nodes over UCT selection.

*(2) Expansion.* The expansion phase generates child nodes from the selected node. Valid actions are determined by the node type (Table 1) and executed via LLM prompting. Each action is sampled $N_{expansion}$ times with temperature $T_{expansion}$, creating $N_{expansion} \times |E_{valid}|$ child nodes, where $|E_{valid}|$ represents the number of valid actions. This sampling approach enhances reasoning diversity.

*(3) Simulation.* A complete simulation process consists of iterative node selection and expansion until reaching a termination node. Throughout the simulation process, all newly expanded child nodes are persistently maintained within the tree structure.

*(4) Backpropagation.* At a termination node ($v_t$), backpropagation begins by evaluating the predicted SQL using the self-supervised reward function from Section 3.2. We identify the action ($A_5$ or $A_6$) that produced the final SQL, then sample $N_{reward}$ SQL queries with temperature $T_{reward}$ to ensure diversity. The reward value $r$ is calculated from the self-consistency score between the execution results of the sampled and predicted SQLs. The process then backtracks from $v_t$ to the root node, updating $Q(v, a)$ and $N(v)$ values for all nodes along the path $v_0 \oplus \cdots \oplus v_t$ using: $Q(v, a) = Q(v, a) + r$, $N(v) = N(v) + 1$. These updated values guide future search directions through the UCT formula in subsequent rollouts.

After $N_{rollout}$ rollouts, we collect all complete trajectories that reach termination nodes, forming a set of candidate SQL reasoning trajectories $T = \{\tau_1, \tau_2, \ldots, \tau_n\}$.

**Final SQL Selection.** To select the optimal trajectory and SQL query from $T$, we leverage the convergent nature of Text-to-SQL: different reasoning paths yield equivalent SQL queries for a given question. We execute all predicted SQL queries and select the one with the highest execution result consistency as the final prediction. This approach differs from previous methods like CHASE-SQL (Pourreza et al.,

2025), which rely on fine-tuned proprietary models such as Gemini-1.5-Flash, requiring extensive domain-specific data.

We show the pseudo-code of Alpha-SQL in Appendix A.1.

**Pruning Strategies.** Alpha-SQL incorporates schema constraints and semantic rules into the search process to prune invalid paths early. A key aspect of our pruning strategy is the elimination of redundant nodes. For instance, when performing a Schema Selection action, we may sample the LLM $M$ multiple times (*e.g.*, 3 times). Although the Chain-of-Thought content generated by $M$ may differ in each sample, if the final selected schema subset is identical, we create only one child node instead of three duplicate nodes. This de-duplication significantly reduces the branching factor of the search tree without loss of information.

### 4.3. Database Value Retrieval

Efficiently and accurately retrieving database values relevant to a user's natural language question is crucial for Text-to-SQL generation (Liu et al., 2024). SQL queries typically reference a limited subset of column values extracted from potentially extensive databases, primarily for utilization within filtering conditions, such as those specified in WHERE clauses. However, a significant challenge arises from the potential semantic discrepancies between how a user phrases their query and how values are stored in the database (*e.g.*, a user might say "America" while the database stores "United States"). Following Talaei et al. (2024), we employ a two-stage approach involving offline pre-processing and online retrieval.

**Offline Pre-processing.** In the offline phase, we focus on pre-processing database values to facilitate fast and accurate retrieval during the online phase. This process specifically targets *TEXT* type column values, as these are most prone to ambiguities and variations in user queries. For these selected textual column values, we apply the MinHash (Datar et al., 2004) technique to generate compact signatures. These MinHash signatures are then stored locally, creating an indexed and optimized representation of the database values that are most likely to require disambiguation.

**Online Retrieval.** During the online phase, when a user submits a natural language question, we first extracts relevant keywords from the question, which can be guided by few-shot prompts (Appendix A.3). Once the keywords are identified, Locality Sensitive Hashing (LSH) (Datar et al., 2004) is utilized in conjunction with the pre-computed MinHash signatures stored locally. This LSH-based search allows for efficient retrieval of database values that are semantically similar to the extracted keywords. The retrieved values are then subject to further filtering based on predefined editing similarity and semantic similarity thresholds ($\epsilon_{edit}$, $\epsilon_{semantic}$). The semantic matching employs OpenAI's *text-embedding-*

*3-large* model. The finally selected, relevant database values are then incorporated into the database schema prompt provided to our LLM-as-Action-Model, enriching the context for the SQL generation process.

## 5. Experiments

### 5.1. Experimental Setup

**Datasets.** We utilize the **Spider** (Yu et al., 2018) and **BIRD** (Li et al., 2023c) development sets for evaluation. Spider contains 1034 (NL, SQL) pairs, and BIRD includes 1534 pairs, with BIRD queries being more complex and containing domain-specific keywords like CASE and IIF.

To facilitate more comparison experiments while reducing computational costs (Sections 5.3 to 5.5), we follow CHESS-SQL (Talaei et al., 2024) and utilize the same Subsampled Development Set (**SDS**), which comprises 10% of each database from the BIRD development set. The SDS contains 147 samples, consisting of 81 simple, 54 moderate, and 12 challenging questions.

**Metrics.** Following prior work (Pourreza et al., 2025), we use Execution Accuracy (EX) as the metric, defined as the percentage of predicted SQL queries that generate execution results identical to those of the ground-truth queries.

**Hardware.** All experiments are run on an Ubuntu 22.04.3 LTS server with 512GB of RAM and dual 40-core Intel(R) Xeon(R) Platinum 8383C CPUs (@ 2.70GHz). Open-source LLMs are deployed locally using 8 GPUs, each with 80GB of memory and 312 TFLOPS with BF16 precision.

### 5.2. Main Results on BIRD and Spider Datasets

**Settings.** We employed three open-source models from the Qwen2.5-Coder family - 7B, 14B, and 32B (Yang et al., 2024) - as inference models for Alpha-SQL. The related hyper-parameters were set as follows: For offline database value retrieval, we set the editing similarity $\epsilon_{edit}$ as 0.3 and semantic similarity $\epsilon_{semantic}$ as 0.6. For the MCTS rollout process, we set the number of rollouts to $N_{rollout} = 24$. During node expansion, each action was sampled $N_{expansion} = 3$ times with a sampling temperature of $T_{expansion} = 0.8$. In the computation of self-supervised rewards, we set the SQL sampling parameters with $N_{reward} = 5$ repetitions and a temperature of $T_{reward} = 1.0$. For the SQL Revision action ($A_6$), we set a maximum iteration limit of $N_{revision} = 10$ for the multi-round correction process.

**Performance on BIRD Dataset.** As shown in Table 2, we conducted a comprehensive comparison between Alpha-SQL and current state-of-the-art approaches, categorizing methods based on their zero-shot capabilities. Alpha-SQL, leveraging the relatively lightweight Qwen2.5-Coder-7B model, achieved 66.8% average accuracy, comparable to the

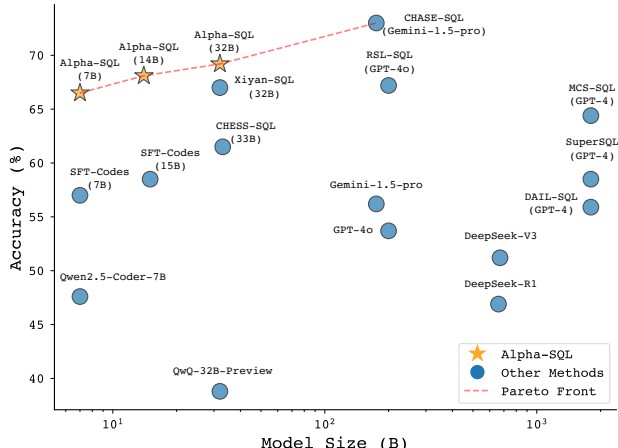

*Figure 4.* Performance vs. Model Size on the BIRD dev. For GPT-4, GPT-4o, and Gemini-1.5-pro, we referenced the parameter descriptions from (Abacha et al., 2024) for plotting.

performance of RSL-SQL (Cao et al., 2024), which relies on the proprietary GPT-4o. Notably, this performance surpasses many methods that require data fine-tuning. Upon scaling our inference model to 32B parameters, Alpha-SQL demonstrated superior performance in the zero-shot scenario, with 69.7% average accuracy. Even when compared to methods with domain data fine-tuning, Alpha-SQL's performance is exceeded only by CHASE-SQL (Pourreza et al., 2025), which requires fine-tuning the proprietary Gemini-1.5-Flash model, and XiYan-SQL (Gao et al., 2024b), which fine-tunes an unknown model. These results confirm Alpha-SQL's effectiveness as a plug-and-play framework that delivers competitive performance without fine-tuning.

**Performance on Spider Dataset.** We also evaluate Alpha-SQL on the Spider development dataset. As shown in Table 3, Alpha-SQL with Qwen2.5-Coder-14B outperforms existing methods. Notably, it achieves a 2.1% improvement over SFT CodeS-15B (Li et al., 2024b), which was specifically fine-tuned for the Spider dataset. This demonstrates Alpha-SQL's capability to achieve strong generalization performance without dataset-specific fine-tuning.

**Performance-Scale Trade-off Analysis.** To explore the performance potential of smaller open-source LLMs, we conducted this experiment to demonstrate that Alpha-SQL can unlock the full potential of smaller models while maintaining cost efficiency. Figure 4 shows that Alpha-SQL significantly outperforms larger models on the Pareto frontier, enabling smaller models, such as the 7B and 14B versions, to achieve accuracy comparable to or surpassing much larger models, including GPT-4o-based approaches. This demonstrates our framework's ability to optimize Text-to-SQL performance across varying model scales.

*Table 2.* Execution Accuracy on BIRD Development Dataset.

| Method | Inference Model | Selection Model | Zero-shot Setting | Accuracy (%) | | | |
|---|---|---|---|---|---|---|---|
| | | | | Simple | Moderate | Challenging | All |
| SFT CodeS (Li et al., 2024b) | CodeS-7B | - | ✗ | 64.6 | 46.9 | 40.3 | 57.0 |
| SFT CodeS (Li et al., 2024b) | CodeS-15B | - | ✗ | 65.8 | 48.8 | 42.4 | 58.5 |
| Distillery (Maamari et al., 2024) | GPT-4o | - | ✗ | - | - | - | 67.2 |
| CHESS-SQL (Talaei et al., 2024) | Deepseek-Coder-33B | GPT-4-Turbo | ✗ | - | - | - | 65.0 |
| CHESS-SQL (Talaei et al., 2024) | Deepseek-Coder-33B | LLaMA3-70B | ✗ | - | - | - | 61.5 |
| CHASE-SQL (Pourreza et al., 2025) | Gemini-1.5-Pro | Gemini-1.5-Flash | ✗ | - | - | - | 73.0 |
| XiYan-SQL (Gao et al., 2024b) | ? | ? | ✗ | - | - | - | 73.3 |
| XiYan-SQL (Gao et al., 2024b) | Qwen2.5-Coder-32B | Qwen2.5-Coder-32B | ✗ | - | - | - | 67.0 |
| DAIL-SQL (Gao et al., 2024a) | GPT-4 | SC Selection | ✓ | 63.0 | 45.6 | 43.1 | 55.9 |
| SuperSQL (Li et al., 2024a) | GPT-4 | SC Selection | ✓ | 66.9 | 46.5 | 43.8 | 58.5 |
| MCS-SQL (Lee et al., 2025) | GPT-4 | GPT-4 | ✓ | - | - | - | 64.4 |
| RSL-SQL (Cao et al., 2024) | GPT-4o | GPT-4o | ✓ | 74.4 | 57.1 | 53.8 | 67.2 |
| **Alpha-SQL (Ours)** | **Qwen2.5-Coder-7B** | **SC Selection** | ✓ | **72.6** | **59.3** | **53.1** | **66.8** |
| **Alpha-SQL (Ours)** | **Qwen2.5-Coder-14B** | **SC Selection** | ✓ | **74.6** | **61.0** | **55.9** | **68.7** |
| **Alpha-SQL (Ours)** | **Qwen2.5-Coder-32B** | **SC Selection** | ✓ | **74.5** | **64.0** | **57.2** | **69.7** |

*Table 3.* Execution Accuracy on Spider Development Dataset.

| Method | Inference Model | Selection Model | Zero-shot Setting | Accuracy (%) | | | | |
|---|---|---|---|---|---|---|---|---|
| | | | | Easy | Medium | Hard | Extra Hard | All |
| SFT CodeS (Li et al., 2024b) | CodeS-7B | - | ✗ | 94.8 | 91.0 | 75.3 | 66.9 | 85.4 |
| SFT CodeS (Li et al., 2024b) | CodeS-15B | - | ✗ | 95.6 | 90.4 | 78.2 | 61.4 | 84.9 |
| C3-SQL (Dong et al., 2023) | GPT-3.5-Turbo | SC Selection | ✓ | 92.7 | 85.2 | 77.6 | 62.0 | 82.0 |
| DIN-SQL (Pourreza & Rafiei, 2023) | GPT-4 | - | ✓ | 92.3 | 87.4 | 76.4 | 62.7 | 82.8 |
| DAIL-SQL (Gao et al., 2024a) | GPT-4 | SC Selection | ✓ | 91.5 | 90.1 | 75.3 | 62.7 | 83.6 |
| ZeroNL2SQL (Fan et al., 2024) | GPT-4 | - | ✓ | - | - | - | - | 84.0 |
| MAC-SQL (Wang et al., 2025) | GPT-4 | - | ✓ | - | - | - | - | 86.8 |
| SuperSQL (Li et al., 2024a) | GPT-4 | SC Selection | ✓ | 94.4 | 91.3 | 83.3 | 68.7 | 87.0 |
| **Alpha-SQL (Ours)** | **Qwen2.5-Coder-7B** | **SC Selection** | ✓ | **94.0** | **89.2** | **76.4** | **63.3** | **84.0** |
| **Alpha-SQL (Ours)** | **Qwen2.5-Coder-14B** | **SC Selection** | ✓ | **94.0** | **91.0** | **79.9** | **72.3** | **87.0** |

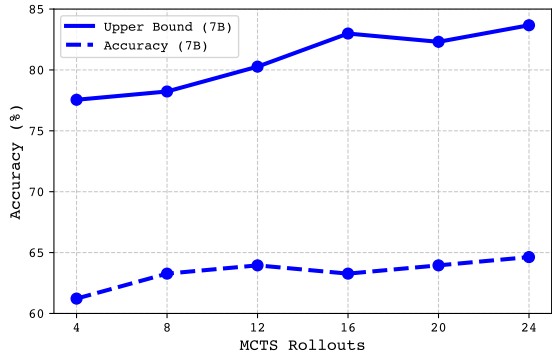

*Figure 5.* Accuracy vs. MCTS Rollouts.

the number of MCTS rollouts affects performance, we conducted an in-depth analysis. We conducted experiments on the SDS dataset using the Qwen2.5-Coder-7B model, maintaining all hyper-parameters identical to Section 5.2 except for the number of rollouts ($N_{rollout}$). We report both the upper bound accuracy and final accuracy. Similar to CHASE-SQL (Pourreza et al., 2025), the upper bound accuracy represents the percentage of samples where the candidate SQL set contains the correct SQL query before the final SQL selection. We observe a positive correlation between the number of MCTS rollouts and both upper bound and final accuracy metrics. This demonstrates Alpha-SQL's capability to enhance Text-to-SQL task performance through more MCTS explorations.

### 5.3. Impact of MCTS Rollouts on Performance

The efficiency of MCTS in exploring large search spaces is a key feature of Alpha-SQL. Based on Table 1, we calculated that there are over 3000 possible reasoning paths for each text-to-SQL task. Remarkably, Alpha-SQL achieves significant performance improvements with just 24 MCTS rollouts, suggesting that our Alpha-SQL can efficiently explore significantly larger search spaces.

To further investigate this efficiency and understand how

### 5.4. Comparison with Baseline LLMs

In this section, we present an evaluation of the baseline performance characteristics of various LLMs when applied directly to the Text-to-SQL task on the SDS dataset. These LLMs were categorized into two groups: general LLMs, exemplified by models such as GPT-4o, and LLMs specifically optimized for reasoning tasks, such as DeepSeek-R1. For a fair comparison, all baseline LLMs were prompted

*Table 4.* Comparison with Baseline LLMs on the SDS dataset.

| Model | Accuracy (%) |
| --- | --- |
| Deepseek-V3 | 51.2 |
| GPT-4o | 53.7 |
| Gemini-1.5-Pro | 56.2 |
| QwQ-32B-Preview | 38.8 |
| DeepSeek-R1 | 50.3 |
| Gemini-2.0-Flash-Thinking-Exp | 60.8 |
| Qwen2.5-Coder-7B | 47.6 |
| **+ Alpha-SQL (Ours)** | **64.6 ($\uparrow$ 17.0)** |
| Phi-4 | 43.5 |
| **+ Alpha-SQL (Ours)** | **60.0 ($\uparrow$ 16.5)** |

*Table 5.* Ablation Study on the Action Space.

| Action Space | Accuracy (%) |
| --- | --- |
| $A_1, A_2, A_3, A_4, A_5, A_6, A_7$ | 64.6 |
| w/o $A_1$ (Question Rephrasing) | 63.9 ($\downarrow$ 0.7) |
| w/o $A_2$ (Schema Selection) | 63.1 ($\downarrow$ 1.5) |
| w/o $A_3$ (Column Value Identification) | 64.2 ($\downarrow$ 0.4) |
| w/o $A_4$ (Column Function Identification) | 64.0 ($\downarrow$ 0.6) |
| w/o $A_6$ (SQL Revision) | 62.8 ($\downarrow$ 1.8) |

using a standardized Text-to-SQL prompt, detailed in Appendix A.4. As shown in Table 4, our Alpha-SQL, utilizing a model with only 7B parameters, surpasses all baseline models in performance. Notably, Alpha-SQL outperforms Gemini-2.0-Flash-Thinking-Exp, a sophisticated reasoning-optimized model, despite using a weaker model. This shows that the Text-to-SQL task requires targeted reasoning optimization, which is a strength of the Alpha-SQL framework. Moreover, to validate Alpha-SQL's plug-and-play advantages, we conducted additional experiments using Phi-4 (Abdin et al., 2024) and Qwen2.5-Coder-7B as inference models. Compared to directly prompting these LLMs, Alpha-SQL achieved significant accuracy improvements of 17.0% and 16.5% for Qwen2.5-Coder-7B and Phi-4, respectively. These results validate Alpha-SQL's generalizability and effectiveness across different inference models.

### 5.5. Ablation Study of Action Space

The purpose of this ablation study is to validate the effectiveness of our proposed Text-to-SQL reasoning action space and the LLM-as-Action-Model approach. To achieve this, we conducted experiments on the SDS dataset, systematically removing individual actions from the original action space while maintaining the parameter settings from Section 5.2. Table 5 presents the results of these experiments.

Table 5 shows that removing any action from the original action space negatively impacts performance. The SQL Revision action demonstrates particular significance, as it leverages database interaction to incorporate feedback into the LLM for SQL correction, highlighting the importance of database execution feedback for Text-to-SQL tasks.

## 6. Conclusion

In this paper, we proposed Alpha-SQL, a zero-shot Text-to-SQL framework that frames SQL generation as a structured search problem. By combining Monte Carlo Tree Search with LLM-as-Action-Model, Alpha-SQL explores the SQL query space efficiently without fine-tuning the LLMs. Experiments show competitive performance, with 69.7% on the BIRD development set. Ablation studies validate the effectiveness of our reasoning actions, and performance improves with more MCTS rollouts.

## Acknowledgements

This paper is supported by NSF of China (62402409, 62436010, and 62441230), Guangdong provincial project 2023CX10X008, Guangdong Basic and Applied Basic Research Foundation (2023A1515110 545), Guangzhou Basic and Applied Basic Research Foundation (2025A04J3935), and Guangzhou-HKUST(GZ) Joint Funding Program (2025A03J3714).

## Impact Statement

This paper presents work whose goal is to advance the field of Machine Learning. There are many potential societal consequences of our work, none of which we feel must be specifically highlighted here.

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

# A. Appendix

## A.1. Alpha-SQL Algorithm

The Alpha-SQL algorithm, as outlined in Algorithm 1, operates in multiple phases: Selection, Expansion, Simulation, and Backpropagation. Given a user query $q$ and the corresponding database schema $\mathcal{D}$, the algorithm starts by initializing an empty search tree $\Psi = (V, E)$ with a root node $v_0$ representing the initial state (lines 3-4). The process then iterates over $N_{rollout}$ MCTS rollouts (line 6), where each rollout seeks to explore high-value reasoning paths in the search space.

In the **Selection phase** (lines 7-12), starting from the root node, Alpha-SQL recursively traverses the search tree until an unexpanded node is reached. The next action $a$ is selected based on the Upper Confidence Bound for Trees (UCT) formula, which balances exploration and exploitation by considering both the estimated reward and the visit counts of actions (line 10). This process continues until a terminal node is found or all children of the node are expanded.

In the **Expansion phase** (lines 13-23), valid next actions $A_{\text{valid}}$ are determined based on the current node's type. The algorithm generates new states by invoking our *LLM-as-Action-Model* to execute each action, creating new child nodes in the search tree (lines 15-21). The expansion process introduces new reasoning paths, enriching the search space for subsequent rollouts.

The **Simulation phase** (lines 24-30) performs rollouts by randomly selecting unexplored child nodes and executing the corresponding actions. This phase continues until a terminal node is reached, at which point the final SQL query is extracted from the trajectory.

Finally, the **Backpropagation phase** (lines 32-40) updates the values of nodes along the path from the terminal node back to the root. The reward is computed by sampling multiple SQL queries and calculating their self-consistency scores based on their execution results. The values of nodes $N(u)$ and $Q(u, a_u)$ are updated according to the reward $r$ to guide future searches.

At the end of the rollouts, the algorithm selects the SQL query with the highest self-consistency from the set of candidate queries generated during the search, as indicated by line 43.

This procedure enables Alpha-SQL to efficiently explore the search space of SQL queries, balancing exploration with accuracy and providing a scalable, fine-tuning-free solution to zero-shot Text-to-SQL tasks.

---

**Algorithm 1** Alpha-SQL: Zero-Shot Text-to-SQL using Monte Carlo Tree Search

---

1: **Input:** Question $q$, Database Schema $D$, LLM $M$, Number of rollouts $N_{rollout}$
2: **Output:** SQL query $y$
3: Initialize an empty search tree $\Psi = (V, E)$ with root node $v_0$
4: Initialize a root node $v_0 \in V$ with $(q, D)$
5: $T \leftarrow \emptyset$
6: **for** $i = 1$ to $N_{rollout}$ **do**
7:     // Selection Phase
8:     $v \leftarrow v_0$
9:     **while** $v$ is not terminal and $v$ is fully expanded **do**
10:       $a \leftarrow argmax_{a \in A(v)}\{Q(v, a)/N(v, a) + c\sqrt{\ln N(v)/N(v, a)}\}$
11:       $v \leftarrow$ child of $v$ reached by action $a$
12:     **end while**
13:     // Expansion Phase
14:     **if** $v$ is not terminal **then**
15:       $A_{valid} \leftarrow$ GetValidActions($v$) from Table 1
16:       **for** $a \in A_{valid}$ **do**
17:         **for** $j = 1$ to $N_{expansion}$ **do**
18:           $v_{new} \leftarrow$ M($q, D, Actions(v_0, \ldots, v), Prompt(a)$)
19:           Add $v_{new}$ as child of $v$ in $\Psi$
20:         **end for**
21:       **end for**
22:       $v \leftarrow$ Random unexplored child of $v$
23:     **end if**
24:     // Simulation Phase
25:     **while** $v$ is not terminal **do**
26:       $A_{valid} \leftarrow$ GetValidActions($v$)
27:       Expand $v$ with $A_{valid}$ and ramdomly select action $a$
28:       $v_{new} \leftarrow$ M($q, D, Actions(v_0, \ldots, v), Prompt(a)$)
29:       $v \leftarrow v_{next}$
30:     **end while**
31:     Extract final SQL $y$ from $v$
32:     // Backpropagation Phase
33:     Sample $N_{reward}$ SQL queries $\{y_1, ..., y_{N_{reward}}\}$ with temperature $T_{reward}$
34:     $r \leftarrow \frac{1}{N_{reward}} \sum_{i=1}^{N_{reward}} \mathbb{1}[\text{Execute}(y, D) = \text{Execute}(y_i, D)]$
35:     $\tau \leftarrow$ the path from $v_0$ to $v$
36:     **for** each node $u$ in $\tau$ **do**
37:       $N(u) \leftarrow N(u) + 1$
38:       $Q(u, a_u) \leftarrow Q(u, a_u) + r$ where $a_u$ is the action taken at $u$
39:       $N(u, a_u) \leftarrow N(u, a_u) + 1$
40:     **end for**
41:     $T \leftarrow T \cup \{\tau\}$
42: **end for**
43: **return** $argmax_{y \in Y}\{\text{Self-consistency}(y)\}$ where $Y$ are unique SQLs from $T$

---

## A.2. Prompt Template for Actions

In this section, we provided the prompt templates for the actions defined in Section 4.1.

---

**Rephrase Question Action Prompt**

You are an AI assistant to help me rephrase questions by splitting the question context into conditions. In your rephrased question, remember to fully express the information in the original question.

Example 1:
Original Question: Name movie titles released in year 1945. Sort the listing by the descending order of movie popularity.
Hint: released in the year 1945 refers to movie_release_year = 1945;
Rephrased Question: Given a list of conditions, please answer the question. Condition 1: Movies are released in the year 1945. Condition 2: Movies are sorted by the descending order of movie popularity. Condition 3: Return the movie titles. Question: What are the movie titles released in the year 1945, sorted by the descending order of movie popularity?

Example 2:
Original Question: How many office supply orders were made by Cindy Stewart in the south superstore?
Hint: office supply refers to Category = 'Office Supplies'
Rephrased Question: Given a list of conditions, please answer the question. Condition 1: Orders are made by Cindy Stewart. Condition 2: Orders are office supplies, refer to Category = 'Office Supplies'. Condition 3: Return the number of orders. Question: How many office supply orders were made by Cindy Stewart in the south superstore?

Example 3:
Original Question: Tell the number of fights landed earlier on Miami Airport on 2018/8/12.
Hint: landed on refers to DEST; landed earlier refers to ARR_DELAY < 0; Miami Airport refers to DEST = 'MIA'; on 2018/8/12 refers to FL_DATE = '2018/8/12';
Rephrased Question: Given a list of conditions, please answer the question. Condition 1: Flights landed on Miami Airport on 2018/8/12, refer to DEST = 'MIA' and FL_DATE = '2018/8/12'. Condition 2: Flights landed earlier, refer to ARR_DELAY < 0. Condition 3: Return the number of fights. Question: How many fights landed earlier on Miami Airport on 2018/8/12?

Answer the following question:
Original Question: {QUESTION}
Hint: {HINT}
Rephrased Question:

---

*Figure 6.* Question Rephrasing Action Prompt.

---

**Schema Selection Action Prompt**

You are an expert and very smart data analyst.

Your task is to examine the provided database schema, understand the posed question, and use the hint to pinpoint the specific columns within tables that are essential for crafting a SQL query to answer the question.

The schema offers an in-depth description of the database's architecture, detailing tables, columns, primary keys, foreign keys, and any pertinent information regarding relationships or constraints. Special attention should be given to the examples listed beside each column, as they directly hint at which columns are relevant to our query.

For key phrases mentioned in the question, we have provided the most similar values within the columns denoted by "– Value Examples" in front of the corresponding column names. This is a critical hint to identify the columns that will be used in the SQL query.

The hint aims to direct your focus towards the specific elements of the database schema that are crucial for answering the question effectively.

Task: Based on the database schema, question, and hint provided, your task is to identify all and only the columns that are essential for crafting a SQL query to answer the question. For each of the selected columns, explain why exactly it is necessary for answering the question. Your reasoning should be concise and clear, demonstrating a logical connection between the columns and the question asked.

Please respond with a JSON object structured as follows:
```json
{ "chain_of_thought_reasoning": "Your reasoning for selecting the columns, be concise and clear.", "table_name1":
["column1", "column2", ...], "table_name2": ["column1", "column2", ...], ... }
```
Make sure your response includes the table names as keys, each associated with a list of column names that are necessary for writing a SQL query to answer the question. For each aspect of the question, provide a clear and concise explanation of your reasoning behind selecting the columns. Take a deep breath and think logically. If you do the task correctly, I will give you 1 million dollars.
Database Schema Overview: {SCHEMA_CONTEXT}
Question: {QUESTION}
Hint: {HINT}
Only output a json (starting with ```json and ending with ```) as your response.

*Figure 7.* Schema Selection Action Prompt.

---

**Column Value Identification Action Prompt**

You are an AI assistant to help me identify the potential column values (if needed to be used in the SQL query) that are essential for answering the question.
Here is an example:
Database Schema:
CREATE TABLE generalinfo
(
id_restaurant INTEGER not null primary key,
food_type TEXT null, – Value Examples: 'thai' | Column Description: the food type
city TEXT null, – Column Description: the city where the restaurant is located in
);
CREATE TABLE location
(
id_restaurant INTEGER not null primary key,
street_name TEXT null, – Value Examples: 'ave', 'san pablo ave', 'pablo ave' | Column Description: the street name of the restaurant
city TEXT null, – Column Description: the city where the restaurant is located in
foreign key (id_restaurant) references generalinfo (id_restaurant) on update cascade on delete cascade
);
Question:
How many Thai restaurants can be found in San Pablo Ave, Albany?
Hint:
Thai restaurant refers to food_type = 'thai'; San Pablo Ave Albany refers to street_name = 'san pablo ave' AND T1.city = 'albany'
Answer:
Since the restaurants are located in Albany, based on the schema information and the hint, I need to use 'location'.'street_name' = 'san pablo ave' AND 'generalinfo'.'city' = 'albany'.
\*\*\*\*\*\*\*\*\*\*\*\*\*\*\*\*\*\*\*\*\*\*\*\*\*\*\*\*
Now, answer the real question, and you need to follow the answer style of the above examples (answer in two sentences).
Database Schema: {SCHEMA_CONTEXT}
Question: {QUESTION}
Hint: {HINT}
Answer:

*Figure 8.* Column Value Identification Action Prompt.

---

**Column Function Identification Action Prompt**

You are an AI assistant to help me identify the potential column functions (if needed to be used in the SQL query) that are essential for answering the question.
Here is an example:
Database Schema:
CREATE TABLE businesses
(
'business_id' INTEGER NOT NULL,
'name' TEXT NOT NULL, – Column Description: the name of the eatery
PRIMARY KEY ('business_id')
);
CREATE TABLE inspections
(
'business_id' INTEGER NOT NULL, – Column Description: the unique id of the business
'score' INTEGER DEFAULT NULL, – Column Description: the inspection score
'date' TEXT NOT NULL, – Value Examples: '2014-01-24'
FOREIGN KEY ('business_id') REFERENCES 'businesses' ('business_id')
);
Question: What are the names of the businesses that passed with conditions in May 2012?
Hint: name of business refers to dba_name; passed with conditions refers to results = 'Pass w/ Conditions'; in May 2012 refers to inspection_date like '2012-05%'
Answer: Since the businesses passed with conditions in May 2012, I should consider a date-related function to filter the 'inspections'.'date' column. I find that column is of type TEXT, so I can use the strftime('%Y-%m', 'inspections'.'date') = '2012-05' to filter the date.
**************************
Now, answer the real question, and you need to follow the answer style of the above examples (answer in two sentences).
Database Schema: {SCHEMA_CONTEXT}
Question: {QUESTION}
Hint: {HINT}
Answer:

---

*Figure 9.* Column Function Identification Action Prompt.

---

**SQL Generation Action Prompt**

You are an experienced database expert. Now you need to generate a SQL query given the database information, a question and some additional information.

The database structure is defined by the following table schemas (comments after '–' provide additional column descriptions). Note that the "Value Examples" are actual values from the column. Some column might contain the values that are directly related to the question. Use it to help you justify which columns to use.

Given the table schema information description and the 'Question'. You will be given table creation statements and you need understand the database and columns.

You will be using a way called "recursive divide-and-conquer approach to SQL query generation from natural language".

Here is a high level description of the steps.

1. **Divide (Decompose Sub-question with Pseudo SQL):** The complex natural language question is recursively broken down into simpler sub-questions. Each sub-question targets a specific piece of information or logic required for the final SQL query.

2. **Conquer (Real SQL for sub-questions):** For each sub-question (and the main question initially), a "pseudo-SQL" fragment is formulated. This pseudo-SQL represents the intended SQL logic but might have placeholders for answers to the decomposed sub-questions.

3. **Combine (Reassemble):** Once all sub-questions are resolved and their corresponding SQL fragments are generated, the process reverses. The SQL fragments are recursively combined by replacing the placeholders in the pseudo-SQL with the actual generated SQL from the lower levels.

4. **Final Output:** This bottom-up assembly culminates in the complete and correct SQL query that answers the original complex question.

Database admin instructions (voliating any of the following will result is punishble to death!):

1. **SELECT Clause:**
- Only select columns mentioned in the user's question.
- Avoid unnecessary columns or values.

2. **Aggregation (MAX/MIN):**
- Always perform JOINs before using MAX() or MIN().

3. **ORDER BY with Distinct Values:**
- Use 'GROUP BY column' before 'ORDER BY column ASC|DESC' to ensure distinct values.

4. **Handling NULLs:**
- If a column may contain NULL values, use 'JOIN' or 'WHERE <column> IS NOT NULL'.

Repeating the question and hint, and generating the SQL with Recursive Divide-and-Conquer, and finally try to simplify the SQL query using 'INNER JOIN' over nested 'SELECT' statements IF POSSIBLE.

Please respond with a JSON object structured as follows:

```json
{ "chain_of_thought_reasoning": "Your detailed reasoning for the SQL query generation, with Recursive Divide-and-Conquer approach.",
"sql_query": "The final SQL query that answers the question."
} ```

\*\*\*\*\*\*\*\*\*\*\*\*\*\*\*\*\*\*\*\*\*\*\*\*\*\*

Table creation statements: {SCHEMA_CONTEXT}

\*\*\*\*\*\*\*\*\*\*\*\*\*\*\*\*\*\*\*\*\*\*\*\*\*\*

Question: {QUESTION}
Hint: {HINT}

\*\*\*\*\*\*\*\*\*\*\*\*\*\*\*\*\*\*\*\*\*\*\*\*\*\*

Only output a json (starting with ```json and ending with ```) as your response.

---

*Figure 10.* SQL Generation Action Prompt.

---

**SQL Revision Action Prompt**

**\*\*Task Description:\*\***
You are an SQL database expert tasked with correcting a SQL query. A previous attempt to run a query did not yield the correct results, either due to errors in execution or because the result returned was empty or unexpected. Your role is to analyze the error based on the provided database schema and the details of the failed execution, and then provide a corrected version of the SQL query.
\*\*Procedure:\*\*
1. Review Database Schema:
- Examine the table creation statements to understand the database structure.
2. Analyze Query Requirements:
- Original Question: Consider what information the query is supposed to retrieve.
- Hint: Use the provided hints to understand the relationships and conditions relevant to the query.
- Executed SQL Query: Review the SQL query that was previously executed and led to an error or incorrect result.
- Execution Result: Analyze the outcome of the executed query to identify why it failed (e.g., syntax errors, incorrect column references, logical mistakes).
3. Correct the Query:
- Modify the SQL query to address the identified issues, ensuring it correctly fetches the requested data according to the database schema and query requirements.

Based on the question, table schemas, the previous query, and the execution result, analyze the result following the procedure, and try to fix the query. You cannot modify the database schema or the question, just output the corrected query.
Please respond with a JSON object structured as follows:
```json
{
"chain_of_thought_reasoning": "Your detailed reasoning for the SQL query revision.",
"sql_query": "The final SQL query that answers the question.",
}
```
\*\*\*\*\*\*\*\*\*\*\*\*\*\*\*\*\*\*\*\*\*\*\*\*\*
Table creation statements: {SCHEMA_CONTEXT}
\*\*\*\*\*\*\*\*\*\*\*\*\*\*\*\*\*\*\*\*\*\*\*\*\*
Question: {QUESTION}
Hint: {HINT}
\*\*\*\*\*\*\*\*\*\*\*\*\*\*\*\*\*\*\*\*\*\*\*\*\*
Only output a json (starting with ```json and ending with ```) as your response.

*Figure 11.* SQL Revision Action Prompt.

## A.3. Prompt Template for Question Keywords Extraction

In this section we provided the prompt template for the question keywords extraction in Section 4.3.

---

**Question Keywords Extraction Prompt**

Objective: Analyze the given question and hint to identify and extract keywords, keyphrases, and named entities. These elements are crucial for understanding the core components of the inquiry and the guidance provided. This process involves recognizing and isolating significant terms and phrases that could be instrumental in formulating searches or queries related to the posed question.

Instructions:
- Read the Question Carefully: Understand the primary focus and specific details of the question. Look for any named entities (such as organizations, locations, etc.), technical terms, and other phrases that encapsulate important aspects of the inquiry.
- Analyze the Hint: The hint is designed to direct attention toward certain elements relevant to answering the question. Extract any keywords, phrases, or named entities that could provide further clarity or direction in formulating an answer.
- List Keyphrases and Entities: Combine your findings from both the question and the hint into a single Python list. This list should contain:
– Keywords: Single words that capture essential aspects of the question or hint.
– Keyphrases: Short phrases or named entities that represent specific concepts, locations, organizations, or other significant details.

Ensure to maintain the original phrasing or terminology used in the question and hint.

Example 1:
Question: "What is the annual revenue of Acme Corp in the United States for 2022?"
Hint: "Focus on financial reports and U.S. market performance for the fiscal year 2022."
["annual revenue", "Acme Corp", "United States", "2022", "financial reports", "U.S. market performance", "fiscal year"]

Example 2:
Question: "In the Winter and Summer Olympics of 1988, which game has the most number of competitors? Find the difference of the number of competitors between the two games."
Hint: "the most number of competitors refer to MAX(COUNT(person_id)); SUBTRACT(COUNT(person_id where games_name = '1988 Summer'), COUNT(person_id where games_name = '1988 Winter'));"
["Winter Olympics", "Summer Olympics", "1988", "1988 Summer", "Summer", "1988 Winter", "Winter", "number of competitors", "difference", "MAX(COUNT(person_id))", "games_name", "person_id"]

Example 3:
Question: "How many Men's 200 Metres Freestyle events did Ian James Thorpe compete in?"
Hint: "Men's 200 Metres Freestyle events refer to event_name = 'Swimming Men's 200 metres Freestyle'; events compete in refers to event_id;"
["Swimming Men's 200 metres Freestyle", "Ian James Thorpe", "Ian", "James", "Thorpe", "compete in", "event_name", "event_id"]

Task: Given the following question and hint, identify and list all relevant keywords, keyphrases, and named entities.
Question: {QUESTION}
Hint: {HINT}
Please provide your findings as a Python list, capturing the essence of both the question and hint through the identified terms and phrases. Only output the Python list, no explanations needed.

---

*Figure 12.* Question Keywords Extraction Prompt.

## A.4. Prompt Template for Baseline LLMs

In this section we provided the prompt template for the directly calling baseline LLMs in Section 5.4

---

**Baseline Text-to-SQL Prompt**

You are an experienced database expert. Now you need to generate a SQL query given the database information, a question and some additional information.

The database structure is defined by the following table schemas (comments after '–' provide additional column descriptions). Note that the "Value Examples" are actual values from the column. Some column might contain the values that are directly related to the question. Use it to help you justify which columns to use.

Given the table schema information description, the 'Question' and 'Hint', you need to generate a SQL query that answers the question.
Please respond the final sql query in the end of response.
\*\*\*\*\*\*\*\*\*\*\*\*\*\*\*\*\*\*\*\*\*\*\*\*\*\*\*
Table creation statements:
{SCHEMA_CONTEXT}
\*\*\*\*\*\*\*\*\*\*\*\*\*\*\*\*\*\*\*\*\*\*\*\*\*\*\*
Question:
{QUESTION}
Hint:
{HINT}
\*\*\*\*\*\*\*\*\*\*\*\*\*\*\*\*\*\*\*\*\*\*\*\*\*\*\*
Output Format:
<think>
Your thinking process.
</think>
<sql>
The final SQL query.
</sql>

---

*Figure 13.* Baseline Text-to-SQL Prompt.

