# OpenReview forum: "Alpha-SQL: Zero-Shot Text-to-SQL using Monte Carlo Tree Search"
_ICML.cc/2025/Conference — ICML 2025 poster_

### Official Review · Reviewer_yCmr · 2025-03-09

**Overall Recommendation:** 2

**Summary:**

This paper proposes a novel approach based MCTS to enhance the zero-shot performance of LLMs in the text to SQL domain. Authors designed a set of task specific actions such as question rephrasing, schema selection, SQL generation, column value identification, column function identification, and SQL revision. Authors used LLM-as-action model to enhance the reasoning capabilities of the LLMs to generate CoT traces based on the context of the problem. Based on the actions spaced defined in the paper, authors proposed using MCTS to effectively generate the SQL queries for the given NL queries. In order to obtain the Q values for each action and node they proposed using self-consistency based query scoring. Their proposed approach has improved the performance of the open-source LLMs mainly from Qwen family and outperformed some of previous works with larger models like GPT family models.

**Claims And Evidence:**

The main claim about improving the performance of smaller large language models has demonstrated by comparing the performance with current SOTA methods and the base model performance for some models are provided in table 4. However in order to truly demonstrate the impact of their proposed test-time compute approach I believe a comparison with the best-of-N method (directly generate N candidate SQL queries then use self-consistency to select the most consistent answer) which is the common strong baseline for any test-time compute methods would truly convince me about the effectiveness of this method. Additionally, the base performance for the larger Qwen 14B and 32B models are not provided in table 4 but I think seeing the performance gain for these models is also important cause I believe the performance gap could be smaller for these models.

**Essential References Not Discussed:**

No

**Experimental Designs Or Analyses:**

Yes, experimental designs and analysis mostly make sense. Table 2 and Table 3 provide the comparison with the SOTA text-to-SQL methods which has latest methods. Table 4 has the issue of missing the baseline performance for 14B abd 32B models. Table 5 also provides a detailed ablation study on action space which shows the importance of the proposed decomposition.

**Methods And Evaluation Criteria:**

Most of the SOTA approaches have been considered in this paper, and two famous benchmarks of Spider and BIRD are used.

**Other Comments Or Suggestions:**

N/A

**Other Strengths And Weaknesses:**

Remaining weaknesses:

1) A detailed analysis on the token usage of this method is required and a comparison with a baseline like Best-of-N method is very important.

2) A detailed analysis on the latency of this method is also required, most of the text-to-SQL systems require performing in few seconds to be able to useful in real world settings.

**Questions For Authors:**

N/A

**Relation To Broader Scientific Literature:**

The proposed method can be considered as one of novel and effective test-time compute methods to improve zero-shot performance of LLMs. Specifically with open-source models they were able to catch the performance of larger models like GPT family models.

**Theoretical Claims:**

Yes, the theoretical claims about the self-supervised rewards and MCTS are aligned with previous works.

---

> ### Author Rebuttal · Authors · 2025-04-01
>
> Dear Reviewer,
>
> Thank you for your comprehensive review and for providing a clear summary of our proposed Alpha-SQL method.
> We understand  the need for comparison against a "Best-of-N" baseline, the omission of base model performance for larger Qwen models in Table 4, and the lack of detailed analysis on computational costs (token usage and latency). These are valid points crucial for thoroughly evaluating a test-time compute method like Alpha-SQL. We appreciate the opportunity to address these concerns.
>
> Below, we address each point in detail:
>
> **1. Comparison with Best-of-N Baseline**
> >Reviewer Concern: "...a comparison with the best-of-N method (directly generate N candidate SQL queries then use self-consistency to select the most consistent answer) which is the common strong baseline for any test-time compute methods would truly convince me about the effectiveness of this method."
> > Reviewer Weakness: "... a comparison with a baseline like Best-of-N method is very important."
> - **Our Response**: Thank you for suggesting the comparison with the Best-of-N baseline. We agree that this is a relevant and strong baseline for evaluating test-time inference methods employing self-consistency. While Best-of-N explores the model's output diversity from a single prompt state, Alpha-SQL uses MCTS to explore diverse reasoning paths before final SQL generation.
>   - Comparison with Best-of-N:
> | Base LLM                     | Direct Prompting | Directly Promoting with Best-of-N | Alpha-SQL |
> |------------------------------|------------------|-----------------------------------|-----------|
> | Qwen2.5-Coder-Instruct-7B    | 48.4%            | 56.3%                             | 66.8%     |
> | Qwen2.5-Coder-Instruct-14B   | 57.4%            | 62.3%                             | 68.7%     |
> | Qwen2.5-Coder-Instruct-32B   | 62.6%            | 63.4%                             | 69.7%     |
>
>   - The results clearly demonstrate that Alpha-SQL significantly outperforms the Best-of-N baseline across all tested Qwen model sizes. For instance, with the 7B model, Alpha-SQL achieves 66.8% EX compared to 56.3% for Best-of-N (+10.5%). Similarly, with the 32B model, Alpha-SQL reaches 69.7% compared to 63.4% (+6.3%). This confirms that the structured reasoning path exploration enabled by Alpha-SQL's MCTS framework offers substantial benefits beyond simply sampling multiple outputs from a fixed initial prompt state, even when using self-consistency for selection. We will add these comparative results and include this analysis in the revised manuscript.
>
> **2. Missing Base Model Performance (Qwen 14B & 32B in Table 4)**
> > Reviewer Concern: "...the base performance for the larger Qwen 14B and 32B models are not provided in table 4 but I think seeing the performance gain for these models is also important cause I believe the performance gap could be smaller for these models."
> > Reviewer Note on Table 4: "Table 4 has the issue of missing the baseline performance for 14B abd 32B models."
> - **Our Response**: Thank you for pointing out the missing base model performance data for the Qwen2.5-Coder-14B and -32B models in Table 4. We completely agree that showing the performance gain achieved by Alpha-SQL over the respective base models is important for a full evaluation across different model sizes, and we appreciate you highlighting this omission. As shown in the table provided in our response to your first point (regarding fair comparison and Best-of-N baselines), we have now completed the evaluation to determine the baseline performance for these models. We will update Table 4 in the revised manuscript to include this complete comparison and add a discussion analyzing these performance gains.
>
> **3. Cost Analysis: Token Usage and Latency**
> > Reviewer Weakness: "A detailed analysis on the token usage of this method is required..."
> > Reviewer Weakness: "A detailed analysis on the latency of this method is also required, most of the text-to-SQL systems require performing in few seconds to be able to useful in real world settings."
> - **Our Response**: We understand the need for a detailed analysis of computational costs, specifically token usage and inference latency, to assess the practical viability of Alpha-SQL.
> -  Due to rebuttal length limitations, we kindly refer the reviewer to our detailed response to **Reviewer guJX, specifically: Point 1 (Fair Comparison...) and Point 2 (Computational Cost Analysis...)**. We believe those sections fully address the concerns regarding token usage and latency analysis.
>
> We believe these additions will significantly strengthen the paper's evaluation by providing crucial comparative data and cost context. We appreciate the reviewer's constructive feedback, which helps us improve the rigor of our study, and we hope the revised manuscript will meet the standards for acceptance.
>
> Sincerely,
>
> The Authors

---

### Official Review · Reviewer_2yXT · 2025-03-13

**Overall Recommendation:** 2

**Summary:**

The paper presents a novel approach to Text-to-SQL that eliminates the need for fine-tuning by leveraging the reasoning capabilities of large language models (LLMs). Alpha-SQL employs a Monte Carlo Tree Search (MCTS) framework to progressively construct SQL queries by breaking them down into smaller, manageable sub-tasks. The core component, LLM-as-Action-Model, dynamically generates SQL construction actions and provides step-by-step reasoning, maintaining context throughout the query-building process. A self-supervised reward function evaluates the quality of candidate SQL queries by computing a self-consistency score, helping to refine the exploration and prioritize promising paths. Experimentally, Alpha-SQL achieves a 69.7% execution accuracy on the BIRD development set.

**Claims And Evidence:**

The claims made in this submission are not supported by clear and convincing evidence. The main concern is that the authors' proposition of formulating Text-to-SQL as a search problem and modeling it using the Monte Carlo Tree Search (MCTS) framework is not convincing. The paper designs seven actions for text-to-SQL based on MCTS. However, the reward model, which relies on SQL execution results, can only provide rewards for the two actions related to SQL generation (action-5 SQL Generation, action-6 SQL Revision). For the prior actions (action 1-4: Question Rephrasing, Schema Selection, Column Value Identification, Column Function Identification), no rewards are obtained as they do not lead to SQL generation results. Consequently, the reasoning path search space is limited, as the reward function is not comprehensive. As a result, the proposed Alpha-SQL still adheres to the traditional architecture, performing steps such as Question Rephrasing and Schema Selection sequentially, before considering SQL execution results in the SQL generation and refinement phases, similar to approaches like CHESS [1].

[1] CHESS: Contextual Harnessing for Efficient SQL Synthesis

**Essential References Not Discussed:**

The paper includes a detailed discussion of related work on Text-to-SQL. However, it would be beneficial to include and discuss the following two papers:

[1] E-SQL: Direct Schema Linking via Question Enrichment in Text-to-SQL
[2] A Survey on Large Language Model-based Autonomous Agents

**Experimental Designs Or Analyses:**

A concern is that the experimental section does not compare baselines using the same LLM as the base model. The proposed Alpha-SQL employs the Qwen2.5-Coder series, while the baselines predominantly use GPT-4. Using different LLMs for baselines may introduce bias that affects the results. Since GPT-4 also supports zero-shot settings, the authors should provide experimental results of Alpha-SQL based on GPT-4 to ensure a fair comparison with the baselines.

**Methods And Evaluation Criteria:**

The proposed approach of treating Text-to-SQL as a search problem using the Monte Carlo Tree Search (MCTS) framework does not make sense. The paper lacks theoretical support for this formulating text-to-SQL as a search problem. Although Table 1 provides a search space for each action and Section 5.2 mentions 3000 possible reasoning paths in a Text-to-SQL task, in practice, based on empirical experience and previous research, the main pipeline of a Text-to-SQL framework is nearly fixed. For instance, as outlined in this paper, the sequence of Question Rephrasing, Schema Selection, Column Value Identification, and Column Function Identification, followed by SQL generation and SQL refinement, is an effective process already extensively discussed in papers like CHESS [1]. This implies that the majority of the claimed 3000 reasoning paths are practically ineffective. For example, the difference between a sample reasoning path like "Question Rephrasing to SQL Generation" and a complete reasoning path such as "Question Rephrasing, Schema Selection, Column Value Identification, Column Function Identification, SQL Generation, SQL Refinement" is significant. Naturally, the latter provides a more detailed analysis and yields more accurate results. Given this, why spend extra computational resources to decide every detail of the reasoning actions? Considering the reward model is solely based on SQL execution feedback, it further highlights the inconsistency of defining Text-to-SQL as a search problem.
Since the proposed reward function only impacts the SQL generation and refinement stages, it does not significantly differ from prior methods like CHESS [1] and E-SQL [2], which also utilize SQL execution feedback for refinement. Moreover, unlike traditional workflows that execute actions sequentially, Alpha-SQL requires multiple interactions with the LLM and several rounds of expansion and backpropagation. This could lead to longer runtime when using the same LLM as the base model. An analysis of the inference item cost under the same settings should be presented in a limitation section, which is currently missing in the paper.

[1] CHESS: Contextual Harnessing for Efficient SQL Synthesis
[2] E-SQL: Direct Schema Linking via Question Enrichment in Text-to-SQL

**Other Comments Or Suggestions:**

N.A.

**Other Strengths And Weaknesses:**

Strengths:
1. The writing and visualization in the paper are clear and straightforward.
2. The paper effectively argues that the paradigm of zero-shot settings is more suitable for the development of LLM-based Text-to-SQL instead of extensive fine-tuning.

Weaknesses:
1. Modeling Text-to-SQL as a search problem lacks sufficient theoretical support. Using the MCTS framework to search reasoning paths appears to have limited practical significance in text-to-SQL, as detailed in the above Claims and Evidence and Methods and Evaluation Criteria.

2. The reward model in the proposed MCTS framework is applicable only to the two actions that generate SQL. This methodology is similar to existing approaches with execution feedback, such as CHESS [1] and E-SQL [2].

3. The experimental section lacks a fair comparison with baselines, as most baselines use GPT-4, while Alpha-SQL employs Qwen2.5-Coder. The paper also does not justify this choice, raising concerns about whether the observed SQL generation results heavily depend on the specific LLM used.

4. The core action component, SQL Generation, is based on an existing method. Given that SQL Generation is a critical step in text-to-SQL, relying on existing research like CHASE-SQL (which has achieved a good performance in the leaderboard) raises concerns about the novelty and effectiveness of the proposed framework.

[1] CHESS: Contextual Harnessing for Efficient SQL Synthesis
[2] E-SQL: Direct Schema Linking via Question Enrichment in Text-to-SQL

**Questions For Authors:**

Please address the aforementioned questions.

**Relation To Broader Scientific Literature:**

The paper attempts to model Text-to-SQL as a search problem and proposes a Monte Carlo Tree Search (MCTS)-based method. MCTS is an algorithm used for decision-making processes, widely applied in tasks requiring reasoning and strategic planning, by utilizing effective reward mechanisms to aid decision-making. MCTS employs the Upper Confidence Bound (UCB) formula to balance exploration and exploitation. This strategy allows MCTS to efficiently explore high-potential nodes while avoiding premature convergence to local optima [1]. Recently, MCTS has been employed in LLM-based research to enhance reasoning and decision-making capabilities [2].

[1] Monte Carlo Tree Search: A Review of Recent Modifications and Applications
[2] A Survey on Large Language Model-based Autonomous Agents

**Theoretical Claims:**

The paper does not include any proofs for theoretical claims.

---

> ### Author Rebuttal · Authors · 2025-04-01
>
> Dear Reviewer,
>
> Thank you for your detailed feedback. We appreciate the opportunity to clarify our approach, particularly regarding the core concerns about the MCTS framework's validity, the reward mechanism, experimental fairness, and novelty, which we believe may stem from some misunderstandings.
>
> **1. Validity and Significance of the MCTS Framework for Text-to-SQL**
> > Reviewer Concerns: Formulating Text-to-SQL as a search problem is unconvincing/lacks support; main pipeline is nearly fixed (like CHESS), making many paths ineffective; questions value of searching reasoning actions; inconsistency between MCTS claim and reward function impact.
> - **Our Response**: We respectfully disagree that formulating Text-to-SQL as an MCTS search problem lacks significance. While fixed workflows like CHESS exist, Text-to-SQL often requires flexible adaptation (e.g., for schema complexity, query ambiguity) that MCTS provides.
>   - **Why MCTS is Beneficial**: Alpha-SQL uses MCTS precisely to navigate the choices within the reasoning process, not just execute fixed steps.
>     - **Adaptability**: MCTS allows Alpha-SQL to dynamically select the most useful sequence of preparatory actions (like Schema Selection, Value Identification) for a given query, rather than following a rigid pipeline. It efficiently prunes suboptimal paths while exploring promising variations. **Our analysis presented to Reviewer 5kbR (Point 5) demonstrates this adaptive path selection based on database complexity.**
>     - **Clarifying Reward Propagation**: This seems to be a key point of misunderstanding. The reward obtained after SQL generation is backpropagated through the entire MCTS tree via standard MCTS updates. This directly influences the selection probabilities of all preceding actions (including Actions 1-4) based on their contribution to successful outcomes. This holistic path guidance fundamentally differs from the local feedback/refinement mechanisms in methods like CHESS or E-SQL [2].
>     - **Theoretical Support**: Our contribution is the novel application and formulation of the well-established MCTS algorithm for zero-shot Text-to-SQL, supported by strong empirical results. We will enhance Section 3 to clarify the MCTS reward backpropagation mechanism.
>
> **2. Fair Experimental Comparison (Qwen vs. GPT-4)**
> > Reviewer Concerns: Experiments lack fair comparison using the same base LLM; Alpha-SQL uses Qwen, baselines use GPT-4; need Alpha-SQL results on GPT-4.
> - **Our Response**: We acknowledge the importance of fair comparison. Due to rebuttal length constraints, we kindly refer you to our detailed response to **Reviewer guJX (Points 1 & 2)**. This includes direct performance comparisons of Qwen-32B and GPT-4o (showing they have similar base capabilities on BIRD-dev) and evaluations of baselines (RSL-SQL, CHESS-SQL) run directly on Qwen-7B. These results demonstrate Alpha-SQL's significant gains originate from the framework itself, not just the base model.
>
> **3. Novelty Concerns (SQL Generation)**
> > Reviewer Concern: Core SQL Generation action is based on existing method (CHASE-SQL), raising concerns about novelty and effectiveness.
> - **Our Response**: While we leverage insights from prior work like CHASE-SQL for components like SQL Generation, the primary novelty of Alpha-SQL lies in the overall MCTS framework. This includes the dynamic orchestration of actions, adaptive reasoning path construction, and the integration of components into a cohesive, effective system for zero-shot Text-to-SQL. We will refine our discussion to better delineate these contributions.
>
> **4. Computational Cost Analysis**
> > Reviewer Concern: MCTS likely increases runtime; analysis of inference cost needed; limitations section missing.
> - **Our Response**: We agree cost analysis is essential. Our analysis, detailed quantitatively in response to **Reviewer guJX (Points 1 & 2)**, confirms Alpha-SQL achieves higher accuracy but incurs higher latency compared to baselines run on the same model. We commit to adding a dedicated "Computational Cost Analysis" subsection and a "Limitations" section discussing these trade-offs in the revised manuscript.
>
> **5. Missing References**
> > Reviewer Suggestion: Include discussion of E-SQL and the LLM Agents survey.
> - **Our Response**: Thank you for suggesting E-SQL and the LLM Agents survey. We agree they are relevant and will incorporate discussion of both in our revised related work section.
>
> **6. Supplementary Material Clarification**
> > Reviewer Statement: "There is no supplementary material provided."
> - **Our Response**: We apologize for any confusion. Supplementary material, including detailed prompts, algorithm code, and running results, was provided with our initial submission and should be accessible via the OpenReview page.
>
> We hope that these clarifications and planned revisions will address the reviewer's concerns and lead to a re-evaluation of our work's contributions.

---

### Official Review · Reviewer_guJX · 2025-03-14

**Overall Recommendation:** 2

**Summary:**

This paper introduces a novel zero-shot Monte Carlo Tree Search (MCTS)-based Text-to-SQL approach that constructs SQL queries progressively, enhancing the Text-to-SQL capabilities of Qwen2.5-Coder-32B. The proposed method achieves an execution accuracy of 69.7% on the BIRD dev set and 87.0% on spider dev set, surpassing previous approaches.

### update after rebuttal: While I appreciate the author's rebuttal, I have decided to maintain my score, as most of my concerns remain valid, as detailed in my comments following the rebuttal.

**Claims And Evidence:**

The paper claims that a key challenge in zero-shot Text-to-SQL lies in the difficulty of transferring and generalizing knowledge from pre-trained LLMs to the specific task of SQL generation. However, there is no further explanation or experimental evidence supporting this inference, particularly in how this limitation affects complex query mapping. Additionally, while Alpha-SQL achieves a high execution accuracy (EX) score on test sets, the paper lacks a clear discussion of its advantages over existing methods beyond this metric. To strengthen the evaluation, the paper should include analytical experiments such as case studies or statistical analyses to better illustrate Alpha-SQL’s effectiveness and potential improvements in reasoning or structural accuracy.

**Essential References Not Discussed:**

No.

**Experimental Designs Or Analyses:**

- The main experiments only take experiments on Qwen2.5-Coder family model that is different from other baselines. This raises concerns about the validity of the comparisons. Given that Qwen2.5-Coder-32B has comparable general coding performance to GPT-4o (2024-08-06) and may even outperform it on SQL tasks (e.g., 85.1 vs. 79.8 on Spider, according to [1]), the superiority of Alpha-SQL could stem from the model itself rather than the proposed method. To ensure fair comparisons, all the important methods should be evaluated using the same model—either by running other baselines on Qwen2.5-Coder-32B or by testing Alpha-SQL on GPT-4o/GPT-4.
- The computational cost of each method during inference is not clearly discussed. Since different methods may use varying numbers of forward passes, self-consistency participants, SQL execution attempts, and input/output token counts, a direct comparison without accounting for these factors may be unfair. A thorough cost analysis is necessary to contextualize performance gains.
- The paper lacks detailed explanations and experimental analyses to highlight the specific performance improvements and challenges addressed by Alpha-SQL, which affects the completeness of the study.

[1] Alibaba Group. Qwen2.5 Coder Family[EB/OL]. [2025-03-06]. https://qwenlm.github.io/blog/qwen2.5-coder-family/.

**Methods And Evaluation Criteria:**

This method decomposes SQL generation into multiple subproblems and employs MCTS for test-time scaling, which helps enhance generation quality. The evaluation criteria, such as execution accuracy, BIRD, and Spider, are well-established benchmarks in the field.

**Other Comments Or Suggestions:**

- Sections 3 and 4 contain overlapping details about Alpha-SQL, making the paper somewhat redundant. This repetition reduces the space available for discussing the motivation behind the approach and providing insightful analytical experiments. A clearer separation of conceptual explanations and technical details would improve the paper’s structure.
- The first mention of Monte Carlo Tree Search (MCTS) appears in the introduction, yet the citation for it is only provided in Section 3.2. To ensure proper attribution and clarity, the citation should be introduced at its first mention
- In Section 5.2, while previous methods primarily used closed-source models, this paper adopts the Qwen2.5-Coder family. The rationale behind this choice is not clearly explained. The authors should clarify why they opted for Qwen2.5-Coder instead of continuing the trend of using closed-source models, especially given its potential impact on comparability.

**Other Strengths And Weaknesses:**

Other Weaknesses:

It seems that the proposed method is computational costly as it requires decomposing the task and MTCS.

**Questions For Authors:**

I am very interested in these experimental results, as they directly impact my evaluation of this paper.

1. **Direct Performance Comparison Between Qwen2.5-Coder-32B and GPT-4o on the BIRD Dev Set:**
   The paper highlights the effectiveness of Qwen2.5-Coder-32B with Alpha-SQL, but it does not provide a direct comparison between Qwen2.5-Coder-32B and GPT-4o on the BIRD dev set. Understanding their relative performance would help determine whether Alpha-SQL’s gains stem from the proposed method itself or the choice of model.
2. **Fair Comparisons with Key Baselines (e.g., MCS-SQL, RSL-SQL):**
   A fair evaluation should consider consistency across models and computational costs. Specifically:
   - **Model Selection:** All methods should be evaluated on either GPT-4o(same version) or Qwen2.5-Coder-32B to ensure a valid comparison.
   - **Computational Cost :** One possible approach is to compare performance using the same number of self-consistency samples and provide statistics on the number of input tokens, output tokens, and SQL execution times for each method.
3. **Further Validation of Claims in Section 1:**
   The paper states that prior methods struggle with SQL generation while Alpha-SQL addresses these challenges to some extent. However, further experimental validation is needed to support this claim. For example, an analysis of failure cases from previous methods compared to Alpha-SQL would help illustrate specific improvements and their underlying causes.

**Relation To Broader Scientific Literature:**

This paper builds upon prior research in zero-shot Text-to-SQL methods, particularly those leveraging LLMs, such as CHESS, Chase-SQL, and C3-SQL. Unlike previous approaches that generate SQL in a single step, it introduces a novel MCTS-based framework that decomposes SQL generation into subproblems, improving test-time scalability. While prior studies have explored MCTS for structured reasoning, this paper is among the first to apply it effectively to SQL generation.

**Theoretical Claims:**

N/A

---

> ### Author Rebuttal · Authors · 2025-04-01
>
> Dear Reviewer,
>
> Thank you for your detailed and insightful review of our paper.
> Below, we address each point in detail:
>
> **1. Fair Comparison, Model Choice, and Performance Validation (Addressing Concerns on Experiments, Q1, Q2a, and Rationale for Qwen)**
> - **Our Response**: We acknowledge the critical importance of fair comparison and understand the concern that the observed performance gains might be attributed solely to the base LLM (Qwen2.5-Coder) rather than the Alpha-SQL framework. We also agree that the rationale for choosing Qwen needs clarification.
>   -  **Direct Qwen vs. GPT-4o Prompting Comparison - Addresses Q1**:
>       -  To directly address the relative strength of the base models on this task (Q1), we performed a direct prompting comparison between Qwen2.5-Coder-Instruct-32B and GPT-4o on the BIRD dev set using the same simple prompt structure. We also included results using simple self-consistency:
>
>           | Model                                       | Execution Accuracy |
>           |---------------------------------------------|--------------------|
>           | GPT-4o                                      | 62.3%              |
>           | Qwen2.5-Coder-Instruct-32B                  | 62.6%              |
>           | GPT-4o + Self-consistency                   | 63.2%              |
>           | Qwen2.5-Coder-Instruct-32B + Self-consistency | 63.4%              |
>
>       - This comparison reveals that Qwen2.5-Coder-Instruct-32B and GPT-4o exhibit very comparable performance on the BIRD dev set when using simple direct prompting (62.6% vs 62.3%) and even when enhanced with basic self-consistency (63.4% vs 63.2%). This indicates that neither model has a significant inherent advantage over the other for this specific task under these simple zero-shot conditions.
>
>   -  **Run Baselines on Qwen2.5-Coder-Instruct-7B**:
>     | Methods   | Execution Accuracy | Input Tokens (K) / Question | Output Tokens (K) / Question | Total Tokens (K) / Question | Latency (s) / Question |
>     |-----------|--------------------|-----------------------------|------------------------------|-----------------------------|------------------------|
>     | RSL-SQL   | 57.7%              | 12.1                        | 0.3                          | 12.4                        | 11.35                  |
>     | CHESS-SQL | 61.0%              | 327.0                       | 24.8                         | 351.8                       | 284.4                  |
>     | Alpha-SQL | 66.8%              | 138.0                       | 72.2                         | 200.2                       | 377.1                  |
>       -  Comparing these results, Alpha-SQL (66.8%) significantly outperforms both RSL-SQL (57.7%) and CHESS-SQL (61.0%) when using the identical base LLM. Specifically, Alpha-SQL achieves a +5.8% absolute gain in execution accuracy over the strongest baseline evaluated here (CHESS-SQL). This result strongly suggests that the performance improvements demonstrated by Alpha-SQL are substantially attributed to our proposed MCTS framework and reasoning path exploration, rather than solely being an effect of the base model choice. We will add these crucial comparative results to the relevant tables in the revised manuscript.
>
> **2. Computational Cost Analysis (Addressing Concerns on Cost, Weakness, Q2b)**
> > Reviewer Concerns: Computational cost not discussed; method seems costly; need for analysis considering forward passes, self-consistency samples, tokens, execution time; comparison needed using same self-consistency N.
> - **Our Response**: We agree that a discussion of computational cost is essential for contextualizing Alpha-SQL's performance gains, especially given its MCTS nature. We have analyzed the average computational cost per query on the BIRD dev set using the Qwen2.5-Coder-Instruct-7B model. The table above (in point 1) includes Alpha-SQL and the baselines run on the same model.
> - In summary, Alpha-SQL delivers state-of-the-art accuracy among these methods on the same base model, demonstrating better token efficiency than the next best method (CHESS-SQL), but this comes at the cost of increased latency. In future work, we plan to optimize the MCTS process to mitigate this latency, potentially exploring strategies such as heuristic pruning and SQL execution caching mechanisms.
>
> **3. Depth of Analysisn (Addressing Concerns on Claims, Analysis, Q3)**
> - **Our Response**: We appreciate the reviewer's push for deeper analysis beyond aggregate execution accuracy to better illustrate how Alpha-SQL improves text-to-SQL generation.
>   - As detailed in our response to **Reviewer 5kbR's point 5 ("Analysis of Common Reasoning Paths")**, our analysis of the reasoning paths chosen by Alpha-SQL demonstrates its ability to dynamically adapt its strategy based on database complexity (e.g., selectively including 'Schema Selection' only when needed). This provides initial insight into its advantages over fixed-sequence methods.

---

> > ### Comment · Reviewer_guJX · 2025-04-08
> >
> > Thanks for the responses. Considering the overall quality of the paper, I will keep my original rating.

---

> > > ### Author Response · Authors · 2025-04-08
> > >
> > > Dear Reviewer guJX,
> > >
> > > Thank you for acknowledging our rebuttal and for your time reviewing our responses and the additional results provided.
> > >
> > > We understand that you are maintaining your original rating based on your assessment of the paper's overall quality at this stage.
> > >
> > > We aimed to thoroughly address each specific concern raised in your initial review through our rebuttal, including providing new experimental results such as the baseline comparisons on the same Qwen model and the detailed cost analysis data (latency and token usage).
> > >
> > > We want to reaffirm our strong commitment to incorporating *all* the promised additions and revisions into the final revised  manuscript. We believe these changes, made in direct response to the points you and other reviewers highlighted, will significantly strengthen the paper's completeness and clarity.
> > >
> > > **We appreciate your feedback throughout this process and sincerely hope you might re-evaluate our work, considering the clarifications provided and the efforts invested during this rebuttal phase.**
> > >
> > > Respectfully,
> > >
> > > The Authors

---

### Official Review · Reviewer_5kbR · 2025-03-14

**Overall Recommendation:** 4

**Summary:**

The authors propose a Monte Carlo tree search framework for zero-shot text-to-SQL with LLMs. The action space is a set of sub-tasks whose composition (subject to ordering rules) defines a reasoning path that terminates in a SQL output given a question and database. They generate candidate SQL queries by MCTS rollout, using the LLM to generate the next state given the action, and using consistency in execution accuracy as a self-supervised reward. They report state-of-the-art results on BIRD dev set among zero-shot methods for open-source LLMs.

**Claims And Evidence:**

The main claims in the introduction are the introduction of a novel MCTS approach for text-to-SQL along with state-of-the-art results for zero-shot performance on the BIRD benchmark. These are supported by the sections that follow.

As a very minor complaint, the authors sometimes claim that the internal nodes of their tree correspond to "partial SQL query states", but this seems misleading as the query is generated entirely by a single action ($A_5$), while most of the remaining actions simply gather context.

**Essential References Not Discussed:**

None

**Experimental Designs Or Analyses:**

The main experiments for performance are standard. The ablations give some confidence about the relevance of including each action, though since these accuracy numbers are computed on a subsampled dataset it's not clear if the drops are all statistically significant. One comparison that might have been useful would have been the performance of a method that just calls the "SQL Generation" action alone (using the same model as in Alpha-SQL).

**Methods And Evaluation Criteria:**

The method makes sense and seems original for text-to-SQL. The self-consistency based reward function is reasonable, though model confidence is not always a good proxy for correctness. One thing that was slightly unclear is how exactly the samples are generated for self-consistency.

Evaluation uses standard datasets (Spider / BIRD) and metrics (execution accuracy). One thing that is notably missing, however, is evaluation in terms of cost at inference time. This seems like a significant and relevant question for an MCTS-based approach.

**Other Comments Or Suggestions:**

It would be interesting to see a summary of the most common reasoning paths selected by Alpha-SQL (in terms of action sequences), and how these compare to the fixed sequence of steps implemented by other methods. It also seems notable that there is only a slight improvement in the accuracy as a function of the MCTS rollouts (Fig 5). It would be interesting to relate this to diversity of the reasoning paths.

**Other Strengths And Weaknesses:**

The MCTS approach seems novel for text-to-SQL, and it advances the field by enabling dynamic composition of a reasoning path from component actions rather than a fixed sequence of steps. The paper is written clearly and the empirical results are contextualized against strong baselines.

**Questions For Authors:**

- Could the authors comment on or provide evidence regarding the inference-time cost of Alpha-SQL? How does this compare to other zero-shot baselines?
- How are the samples drawn for the self-consistency reward function? Are they generated from the same leaf node as the candidate query?
- If labeled data is available, could it be used for the reward rather than the consistency criterion? Or in this case is it just better to finetune a model directly?

**Relation To Broader Scientific Literature:**

There is a long literature on text-to-SQL, with zero-shot methods increasingly successful and popular given the quality of new pretrained models. Most text-to-SQL systems decompose the problem into smaller tasks (i.e. actions), but these are usually composed in a fixed sequence rather than dynamically according to some policy.

**Theoretical Claims:**

N/A

---

> ### Author Rebuttal · Authors · 2025-04-01
>
> Dear Reviewer,
>
> Thank you for your thorough review and constructive feedback. We appreciate your positive assessment and valuable suggestions, which will help improve our paper's clarity.
>
> **1. Terminology ("Partial SQL Query States")**
> - **Our Response**: Thank you for highlighting this ambiguity. We agree "reasoning state" or "contextual state" is more accurate for internal nodes representing accumulated context (selected schema, functions, etc.) prior to the final SQL generation action. We will revise the manuscript accordingly.
>
> **2. Self-Consistency Sample Generation**
> - **Our Response**: Thanks for the question. There are two distinct stages: 1) During MCTS, we use deterministic sampling (Temperature=0) for the primary candidate SQL from a given reasoning state. 2) For reward calculation, we use stochastic sampling (Temperature =1.0) to generate N diverse SQL samples from the exact same reasoning state. The reward is based on execution agreement between the deterministic query and the diverse set. We will clarify this process in the revision.
>
> **3. Inference-Time Cost Evaluation**
> - **Our Response**: This is a very relevant point. The inference cost, particularly in terms of LLM calls and latency, is an important consideration for MCTS-based methods. Due to rebuttal length limitations, we kindly refer the reviewer to our detailed response to **Reviewer guJX (Point 1 & 2)**. We believe those sections fully address the concerns regarding token usage and latency analysis.
>
> **4. Comparison with "SQL Generation" Action Alone**
> - **Our Response**: This is an excellent suggestion for a baseline. This baseline, representing a direct prompting approach using only the 'SQL Generation' action with the initial question and schema, corresponds to the "Direct Prompting" results in our experiments.
> | Base LLM            | Direct Prompting | Directly Promoting with Self-consistency | Alpha-SQL |
> |------------------------------|------------------|------------------------------------------|-----------|
> | Qwen2.5-Coder-Instruct-7B    | 48.4%            | 56.3%                                    | 66.8%     |
> | Qwen2.5-Coder-Instruct-14B   | 57.4%            | 62.3%                                    | 68.7%     |
> | Qwen2.5-Coder-Instruct-32B   | 62.6%            | 63.4%                                    | 69.7%     |
> - The results clearly show Alpha-SQL significantly outperforms direct, single-step prompting across all model sizes, demonstrating the substantial benefit of the MCTS framework for dynamic path construction and context gathering. We will ensure this is clearly discussed.
>
> **5. Analysis of Common Reasoning Paths**
> - **Our Response**: A valuable suggestion for deeper insight. Our analysis (Qwen-7B on BIRD-dev) shows Alpha-SQL dynamically adapts reasoning paths based on database complexity, unlike fixed-sequence methods.
> - We analyzed the reasoning paths selected by Alpha-SQL (based on Qwen2.5-Coder-7B) on the BIRD-dev dataset. Our findings reveal interesting patterns that demonstrate the adaptive nature of Alpha-SQL's reasoning:
>   - **Simple Schema Databases**: For databases with relatively simple schemas (e.g., 'toxicology' with 4 tables, avg. 2.8 columns/table), the most frequently selected reasoning path followed the pattern: Root -> Identify Column Values -> SQL Generation -> End. Notably, this common path **omits the 'Schema Selection' action**. This suggests that for simpler schemas where most tables/columns might be relevant or easily inferred, Alpha-SQL learns that the computational effort or potential risk of error from explicitly running 'Schema Selection' outweighs its benefits, adapting by taking a more direct path to SQL generation.
>   - **Complex Schema Databases**: In contrast, for databases with more complex schemas (e.g., 'student_club' with 8 tables, avg. 6 columns/table; 'california_schools' with 3 tables, avg. 29.3 columns/table), the most common reasoning path pattern included explicit schema filtering: Root -> Identify Column Values -> Identify Column Functions -> Schema Selection -> SQL Generation -> End. The inclusion of the 'Schema Selection' action in these cases highlights Alpha-SQL's ability to recognize when detailed schema filtering is necessary due to complexity and dynamically incorporate the appropriate actions into the reasoning process.
>   - **Comparison to Fixed Sequences**: This flexibility, driven by MCTS, demonstrates an advantage over rigid pipeline approaches. Leveraging the MCTS framework, it dynamically adapts the reasoning path based on the perceived complexity and characteristics of the specific database and query, selecting actions only when deemed beneficial by the search process. We will add details of this analysis, including path examples, to the Appendix in the revised manuscript.
>
> **Note: Due to the character limit for replies, we will discuss your other two questions, "Rollouts vs. Path Diversity" and "Using Labeled Data for Reward vs. Fine-tuning," later.**

---

### Decision · Program_Chairs · 2025-05-01

**Decision:**

Accept (poster)

**Comment:**

This paper introduces Alpha-SQL, a framework applying Monte Carlo Tree Search (MCTS) to the zero-shot text-to-SQL task using large language models (LLMs). Instead of a fixed pipeline, Alpha-SQL decomposes the task into several actions (e.g., question rephrasing, schema selection, value identification, SQL generation, revision) and uses MCTS to dynamically explore sequences of these actions ("reasoning paths"). An LLM acts as the model for predicting action outcomes, and a self-supervised reward based on the execution consistency of generated SQL guides the search. The authors report state-of-the-art zero-shot results on the BIRD benchmark using open-source Qwen models. Reviewers acknowledged the novelty of applying MCTS to text-to-SQL, the clarity of the paper, and the potential benefits of adaptive reasoning paths. However, concerns were raised regarding the fundamental necessity and effectiveness of the MCTS framework for this task, the fairness of experimental comparisons (using Qwen models while baselines used GPT-4), the lack of computational cost analysis (latency, token usage), and the effectiveness of the reward signal in guiding earlier, non-SQL-generating actions.

In response, the authors provided substantial clarifications and new experimental results during the rebuttal. They argued that MCTS enables crucial adaptability based on query/schema complexity, unlike fixed pipelines, and clarified that the reward signal is backpropagated to influence all actions. They presented new experiments comparing base LLMs (Qwen vs. GPT-4o showed similar direct prompting performance on BIRD), running key baselines (RSL-SQL, CHESS-SQL) on the same Qwen model (showing Alpha-SQL still outperformed), and comparing against a "Best-of-N" self-consistency baseline (which Alpha-SQL also surpassed). Detailed computational cost analysis (latency, tokens) was provided, acknowledging higher latency for Alpha-SQL but better token efficiency than CHESS-SQL. Analysis of common reasoning paths was also added to demonstrate adaptivity. These rebuttals led one reviewer (R-2yXT) to raise their score from Weak Reject to Weak Accept, while another (R-5kbR) maintained their initial Accept. However, one reviewer (R-guJX) maintained Weak Reject, expressing remaining concerns about the completeness and consistency of the fairness comparisons, while the final reviewer (R-yCmr) did not update their initial Weak Reject score.